# Guanylate binding proteins directly attack *Toxoplasma gondii* via supramolecular complexes

**Elisabeth Kravets**[1†], **Daniel Degrandi**[1†], **Qijun Ma**[2†], **Thomas-Otavio Peulen**[2], **Verena Klümpers**[1‡], **Suren Felekyan**[2], **Ralf Kühnemuth**[2], **Stefanie Weidtkamp-Peters**[3], **Claus AM Seidel**[2*], **Klaus Pfeffer**[1*†]

[1]Institute of Medical Microbiology and Hospital Hygiene, Heinrich-Heine University Düsseldorf, Düsseldorf, Germany; [2]Institute for Molecular Physical Chemistry, Heinrich-Heine-University Düsseldorf, Düsseldorf, Germany; [3]Center of Advanced Imaging, Heinrich-Heine-University Düsseldorf, Düsseldorf, Germany

**\*For correspondence:** cseidel@ hhu.de (CAS); Klaus.Pfeffer@hhu. de (KP)

[†]These authors contributed equally to this work

**Present address:** [‡]Institute of Transfusion Medicine and Immunology, Medical Faculty Mannheim, Heidelberg University, Heidelberg, Germany

**Competing interests:** The authors declare that no competing interests exist.

**Abstract** GBPs are essential for immunity against intracellular pathogens, especially for *Toxoplasma gondii* control. Here, the molecular interactions of murine GBPs (mGBP1/2/3/5/6), homo- and hetero-multimerization properties of mGBP2 and its function in parasite killing were investigated by mutational, Multiparameter Fluorescence Image Spectroscopy, and live cell microscopy methodologies. Control of *T. gondii* replication by mGBP2 requires GTP hydrolysis and isoprenylation thus, enabling reversible oligomerization in vesicle-like structures. mGBP2 undergoes structural transitions between monomeric, dimeric and oligomeric states visualized by quantitative FRET analysis. mGBPs reside in at least two discrete subcellular reservoirs and attack the parasitophorous vacuole membrane (PVM) as orchestrated, supramolecular complexes forming large, densely packed multimers comprising up to several thousand monomers. This dramatic mGBP enrichment results in the loss of PVM integrity, followed by a direct assault of mGBP2 upon the plasma membrane of the parasite. These discoveries provide vital dynamic and molecular perceptions into cell-autonomous immunity.

## Introduction

IFNγ is an immunomodulatory cytokine that rapidly activates potent host cell effector mechanisms to confront a variety of intracellular pathogens (*Decker et al., 2002*). Some of the most abundantly IFNγ induced proteins are the 65-kDa guanylate-binding proteins (GBPs), which mediate cell-autonomous immunity (*MacMicking, 2012*; *Degrandi et al., 2013*; *Pilla et al., 2014*; *Meunier et al., 2015*). GBPs are related to the dynamin super family of GTPases (*Praefcke and McMahon, 2004*) and are highly conserved throughout the vertebrate lineage (*Vestal and Jeyaratnam, 2011*). The human genome harbors seven GBPs and at least one pseudogene, whereas the mouse genome contains 11 GBPs and two pseudogenes (*Kresse et al., 2008*; *Olszewski et al., 2006*). The gene loci of murine GBPs (mGBPs) are tandemly organized in clusters on chromosomes 3 and 5 (*Degrandi et al., 2007*; *Kresse et al., 2008*).

GBPs contain a conserved GTPase-domain which binds guanine nucleotides with low affinities. This induces nucleotide dependent GBP multimerization and cooperative hydrolysis of GTP via GDP to GMP (*Praefcke et al., 2004*; *Ghosh et al., 2006*; *Kravets et al., 2012*; *Prakash et al., 2000b*). Some GBPs are isoprenylated, endowing them with the ability to associate with intracellular membranous compartments (*Vestal et al., 2000*; *Degrandi et al., 2013*).

**eLife digest** A microscopic parasite called *Toxoplasma gondii* causes a serious disease known as toxoplasmosis in humans and other mammals. Once inside the body, the parasite can infect host cells, where it hides inside a cell structure called a vacuole. However, this triggers self-defense mechanisms in the infected cells that help to control the spread of the parasite in the body. Proteins called guanylate binding proteins – which are normally found as small units in healthy host cells – bind to each other and form larger "complexes" that promote immune responses in that particular cell. However, it was not known how the guanylate binding proteins congregate to form the complexes, or how this activates the cell's defenses.

Here, Kravets et al. use sophisticated fluorescence microscopy techniques with living cells to study the roles of guanylate binding proteins in immune responses during *T. gondii* infection. The experiments show that the proteins are stored as larger units in structures within healthy cells that allow them to relocate quickly to the vacuole when the parasite is detected. Once there, the guanylate binding proteins form large complexes that can contain thousands of protein units. The process requires energy that is released from the break down of a molecule called GTP, and specific chemical modifications to the guanylate binding proteins to allow them to bind to each other.

Further experiments found that the guanylate binding proteins in the complexes assist in weakening the structure of the vacuoles, and that subsequently, one type of protein – called GBP2 – directly attacks the parasite itself. Kravets et al.'s findings set the stage for the development of new therapies that help to fight *T. gondii* infections.

Murine GBPs (mGBPs) exert a major impact on cell-autonomous restriction of *Toxoplasma gondii* (*Yamamoto et al., 2012*; *Degrandi et al., 2007*; *Selleck et al., 2013*; *Degrandi et al., 2013*). *T. gondii* is an apicomplexan protozoan parasite with a broad host range, is distributed worldwide and causes serious and often fatal infections in immunocompromised hosts (*Gazzinelli et al., 2014*). *T. gondii* infection experiments in mice deficient for a cluster of mGBPs on chromosome 3 (*Yamamoto et al., 2012*) or solely for mGBP1 or mGBP2 (*Degrandi et al., 2013*; *Selleck et al., 2013*) prove that mGBPs are essential immune effector molecules mediating antiparasitic resistance. In several cell types distinct mGBPs accumulate at the parasitophorous vacuole membrane (PVM) of *T. gondii* (*Degrandi et al., 2007*; *Kravets et al., 2012*; *Degrandi et al., 2013*).

In previous studies, introduction of point mutations into the key positions of the conserved motifs of the GTPase-domain (R48A, K51A, E99A, D182N) and the isoprenylation site of mGBP2 (C586S), clearly showed that nucleotide binding, multimerization, GTP-hydrolysis and membrane anchoring, are essential for localization in vesicle-like structures (VLS) and for the recruitment of mGBP2 to the PVM of *T. gondii* (*Kravets et al., 2012*; *Degrandi et al., 2013*). However, the assembly of homo- and hetero-mGBP multimers, their composition in distinct subcellular compartments, localization-dependent multimerization as well as their requirement for replication control of *T. gondii* in living cells remained enigmatic.

Therefore quantitative live-cell-imaging technologies were employed revealing seminal information on localization, interaction, concentration, structure and dynamics of biomolecules. To investigate the structure, composition and interaction of proteins, Förster resonance energy transfer (FRET) (*Giepmans et al., 2006*) is combined with Multiparameter fluorescence image spectroscopy (MFIS) (*Kudryavtsev et al., 2007*; *Weidtkamp-Peters et al., 2009*), which enables unique advances in FRET imaging. In MFIS, a variety of fluorescence parameters is monitored simultaneously with pico-second accuracy, allowing the determination of many fluorescence parameters in a *pixel-wise* analysis such as number of photons, anisotropies, fluorescence lifetimes, and signal ratios by statistically most efficient estimators (*Sisamakis et al., 2010*) and to plot distinct parameters in MFIS pixel frequency histograms. The combination of MFIS and FRET experiments (MFIS-FRET) enables a quantitative analysis of the biophysical properties of homomeric and heteromeric molecular complexes in living cells (*Stahl et al., 2013*). This allows the identification and selection of pixel populations with unique properties for a detailed *pixel-integrated* analysis. Importantly, live cell measurements with

MFIS can achieve the resolution and precision of traditional in vitro measurements of molecule ensembles with respect to the number of resolved species and rate constants.

Here, by advanced biophysical MFIS-FRET technology, it is demonstrated that the GTPase activity and isoprenylation of mGBP2 are prerequisites for its multimerization. The multimerization is essential for control of *T. gondii* replication. Colocalization and MFIS analysis of mGBPs showed intermolecular interaction of mGBP2 with itself, with mGBP1 and mGBP3, but not with mGBP6 in VLS in living cells. Interestingly, the interaction partnerships were recapitulated at the PVM of *T. gondii*. Moreover, characteristic interaction affinities of mGBP complexes were individually quantified. For the first time, we show that in the process of attacking *T. gondii*, mGBP2 directly targets the plasma membrane of the parasite after disruption and permeabilization of the PVM. These investigations enable a discrete understanding of the dynamics and intracellular interactions of mGBP effector molecules in *T. gondii* host defense.

## Results

### Multimerization of mGBP2 WT and mutants, determined by intracellular homo-FRET MFIS analysis

Site-directed mutagenesis of mGBP2 revealed that GTP-binding and hydrolysis as well as C-terminal isoprenylation affect the localization of mGBP2 in the cell (*Degrandi et al., 2013*; *Kravets et al., 2012*). However, the role of the GTPase activity and isoprenylation on the multimerization ability of mGBP2 in living cells is unknown.

Therefore, MFIS-FRET measurements and fluorescence-anisotropy-based homo-FRET analysis were employed in living IFN-γ stimulated mGBP2$^{-/-}$ MEFs reconstituted either with GFP-fused mGBP2 WT protein (hereafter referred to as G-mGBP2 MEFs) or with one of the GTPase-domain mutants (R48A, K51A, E99A, D182N) or with the isoprenylation mutant (C586S) (*Figure 1a*).

The mean steady-state anisotropy of GFP in the cytosol was experimentally determined as $<r_D>_{cytosol}$ = 0.328, which is in agreement with the value predicted by the Perrin equation (*Lakowicz, 2006*) using the known mean global rotational diffusion time $\rho_{global} \approx 15$ ns for freely diffusing GFP. When GFP is fused to mGBP2, two opposing effects need to be considered (*Figure 1b*). First, its rotational freedom is restricted and therefore $r_D$ increases; second, homo-FRET between G-mGBP2 complexes reduces $r_D$ by depolarization of the total GFP signal. Consequently, the average steady-state anisotropy of WT G-mGBP2 in the cytosol $<r_D>_{cytosol}$ remained comparable to the value for free GFP (*Figure 1c*). In contrast, the GFP signal intensity ($S_{G,G}$) in VLS increased significantly, indicating an enrichment of mGBP2 molcules in these structures (*Figure 1c*) accompanied by a significant reduction of the average anisotropy $<r_D>_{VLS}$, suggesting an increased mGBP2 homo-multimerization (*Figure 1a,c*).

The nucleotide binding and hydrolysis impaired K51A mutant does not localize in VLS (*Kravets et al., 2012*). This mutant showed a higher average anisotropy ($<r_D>_{cytosol}$ = 0.336) as compared to the cytosolic WT mGBP2 (*Figure 1a,c*) due to the absence of homo-FRET, proving its incapability to form multimers. Next, the mean anisotropies of averages over whole MEFs $<r_D>_{cell}$ were determined (*Figure 1d*). The hydrolytically impaired mGBP2 mutants R48A and E99A (*Kravets et al., 2012*) showed significantly increased $<r_D>_{cell}$ values (*Figure 1a,d*), further proving that the GTPase activity is essential for multimerization in living cells. The nucleotide binding deficient mGBP2 mutant D182N showed significantly increased $<r_D>_{cell}$ value (*Figure 1a,d*) as compared to WT mGBP2 and mutants R48A and E99A reflects the low multimerization capability of this mutant. The recombinant isoprenylation mutant (C586S) did not show altered nucleotide binding, hydrolysis activity or multimerization of mGBP2 in cell-free analyses (*Figure 1—figure supplement 1*, *Table 1*). Nevertheless, this mutant did not localize in VLS (*Degrandi et al., 2013*) and showed anisotropy values comparable to the dysfunctional K51A mutant (*Figure 1d*).

Altogether, these data provide compelling evidence that nucleotide binding and membrane anchoring are prerequisites for multimerization of mGBP2 in living cells. The degree of multimerization of mGBP2 increases from cytosol to VLS.

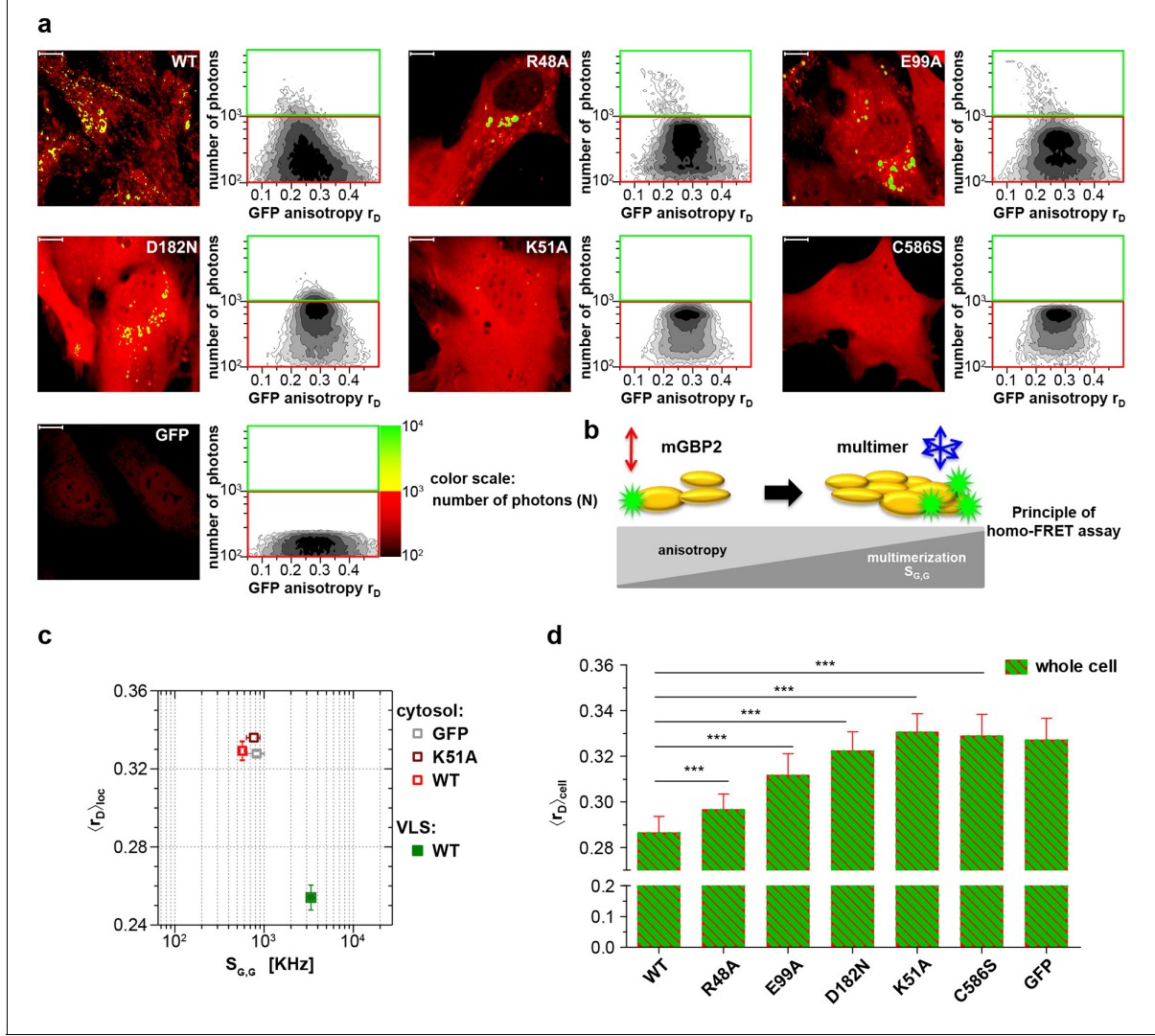

**Figure 1.** Intracellular homo-multimerization of WT and mutant mGBP2. All cells were pre-treated with IFNγ for 16 hr prior investigation (a) Left panel. GFP fluorescence intensity ($S_{G,G}$) images of GBP2$^{-/-}$ MEFs expressing G-mGBP2-WT (G-mGBP2 MEFs), mutants (R48A, K51A, E99A, D182N, C586S) or GFP highlighted with selections of pixels within different cellular compartments. Right panel. MFIS 2D-histograms of GFP anisotropy ($r_D$) on x axis vs. photon number per pixel on y axis, the frequency of pixels color coded from white (lowest) to black (highest). This allows the identification and selection of pixel populations with unique fluorescence properties for a detailed subsequent pixel integrated analysis. The pixels with low photon numbers (below 1000) are selected in red boxes (defined as cytosol) and those with more than 1000 photons in green boxes (defined as VLS). Bars, 10 μm. (b) Scheme of the principle of homo-FRET assays. Compared to G-mGBP2 monomers, $r_D$ in G-mGBP2 multimers decreases due to depolarization of GFP fluorescence while GFP $S_{G,G}$ increases. (c) For specific compartments (cytosol and VLS, respectively), the anisotropy values are averaged over all cells generally denoted as $<r_D>_{loc}$. $<r_D>_{loc}$ and $S_{G,G}$ in cytosol and VLS were plotted for G-mGBP2-WT, and the K51A mutant and GFP in the cytosol. (d) Mean anisotropy of averages over whole cells $<r_D>_{cell}$ for G-mGBP2 WT and mutant proteins. GFP expressing cells served as controls (***p<0.0001).

The following figure supplement is available for figure 1:

**Figure supplement 1.** Biochemical properties and intracellular localization of the C586S mutant of mGBP2.

## Multimerization of mGBP2 WT and mutants at the PVM of *T. gondii*

mGBPs were reported to be involved in rupture of *T. gondii* PVMs few hours after infection and are important for *T. gondii* control in vivo (*Degrandi et al., 2013*; *Selleck et al., 2013*; *Yamamoto et al., 2012*). Previously, it could be determined that the GTPase activity as well as iso-prenylation regulate the recruitment of mGBP2 to the PVM of *T. gondii* (*Degrandi et al., 2013*; *Kravets et al., 2012*). The next step therefore was to elucidate the impact of the GTPase activity

**Table 1.** Dissociation constants $K_D$ of mant-nucleotides for mGBP2 WT and C586S mutant determined by fluorescence titrations and GTPase activity parameters obtained by protein concentration-dependent hydrolysis.

| | Nucleotide binding | | | | | |
|---|---|---|---|---|---|---|
| | mant-GTPγS | mant-GDP | mant-GMP | GTP-hydrolysis | | |
| | $K_D$ (µM) | $K_D$ (µM) | $K_D$ (µM) | $K_{max}$ (min⁻¹) | Dimer $K_D$ (µM) | GMP (%) |
| WT | 0.45 | 0.54 | 14.4 | 102 | 0.029 | 74 |
| C586S | 0.50 | 0.45 | 15.5 | 133 | 0.026 | 72 |

The % GMP indicates the relative amount of the two products, GDP and GMP

and the isoprenylation of mGBP2 on the ability to multimerize at the PVM and to control intracellular *T. gondii* replication. Hence, G-mGBP2 MEFs as well as MEFs expressing GTPase and isoprenylation mutants were infected with *T. gondii* and analyzed by MFIS homo-FRET assays. Also, the ratio of replicative units, so called rosettes, versus single parasites was determined 32 hr after infection (*Figure 2*).

A marked decrease of fluorescence intensities of WT mGBP2 in the cytosol of infected cells (*Figure 2a,b*) compared to uninfected cells (*Figure 1c*) concurrent with a strong increase of the mGBP2 concentration at the PVM of *T. gondii* was observed along with a further decrease in anisotropy (*Figure 2a,b*; *Figure 2—figure supplement 1*). This raises the question on a distinct composition of the mGBP2 complexes at the PVM, which will be addressed below by pixel-integrated MFIS analysis.

As shown previously, the enzymatically dysfunctional K51A and the isoprenylation C586S mutants showed nearly no recruitment to the PVM (*Kravets et al., 2012*; *Degrandi et al., 2013*). Interestingly, as shown here, the corresponding anisotropies (*Figure 2a–c*) did not significantly change in comparison to the uninfected situation (*Figure 1*). These mutants were incapacitated in controlling *T. gondii* replication (*Figure 2d*). The R48A and E99A mutants, which have reduced capacity to recruit to the PVM (*Kravets et al., 2012*), showed slightly increased anisotropy at the PVM as compared to WT mGBP2 (*Figure 2c*) and a reduced capability to restrict *T. gondii* growth (*Figure 2d*). For the D182N mutant a higher anisotropy at the PVM in comparison to WT mGBP2 could be determined, suggesting a lower degree of multimerization. This correlated with insufficient control of *T. gondii* growth, comparable to the K51A and C586S mutants (*Figure 2d*).

In summary, it can be concluded that at the PVM the enrichment of mGBP2 is increased compared to VLS. Nucleotide binding, GTPase activity as well as membrane anchoring regulate the multimerization capability of mGBP2 at the PVM and are prerequisites for the control of *T. gondii* replication.

## Colocalization and hetero-FRET studies of mGBPs

Several members of the mGBP family localize in VLS in IFNγ stimulated cells (*Degrandi et al., 2007*). However, it is unclear whether co-compartmentalization of mGBPs and molecular interactions between them in VLS occur. For this purpose, G-mGBP2 MEFs were cotransduced with mCherry fusion proteins of mGBP1, mGBP2, mGBP3, mGBP5, and mGBP6 (hereafter referred to as G-mGBP2/mCh-mGBPx) and confocal imaging studies were performed. (*Figure 3*, *Figure 3—figure supplement 1*). All of the analyzed mGBPs showed a vesicular distribution except for mGBP5 (*Figure 3*). A correlation of localization could be computed employing the Pearson´s coefficient, P. G-mGBP2/mCh-mGBP2 MEFs showed the most pronounced colocalization indicating that the fluorescence tags do not affect protein localization (P = 0.758 ± 0.093). Confocal images revealed a high correlation of G-mGBP2 positive VLS with mCh-mGBP1 (P = 0.516 ± 0.132) and mCh-mGBP3 VLS (P = 0.65 ± 0.121). mCh-mGBP5 (P = 0.108 ± 0.104) and mCh-mGBP6 (P = 0.338 ± 0.126) scarcely overlapped with G-mGBP2. Thus, the subcellular reservoir of mGBP1, mGBP2 and mGBP3 differed from mGBP6, whereas mGBP5 showed no compartmentalization.

To elucidate whether the colocalization of mGBPs is due to specific protein interactions, MFIS-hetero-FRET measurements were performed using G-mGBP2 as donor and mCh-mGBPx as acceptors (*Figure 4*). In the FRET analysis GFP and mCherry fluorescence intensities ($F_G$ and $F_R$) and the mean fluorescence-weighted donor lifetime $<\tau_D>_f$ were determined for each pixel (*Figure 4a*). By

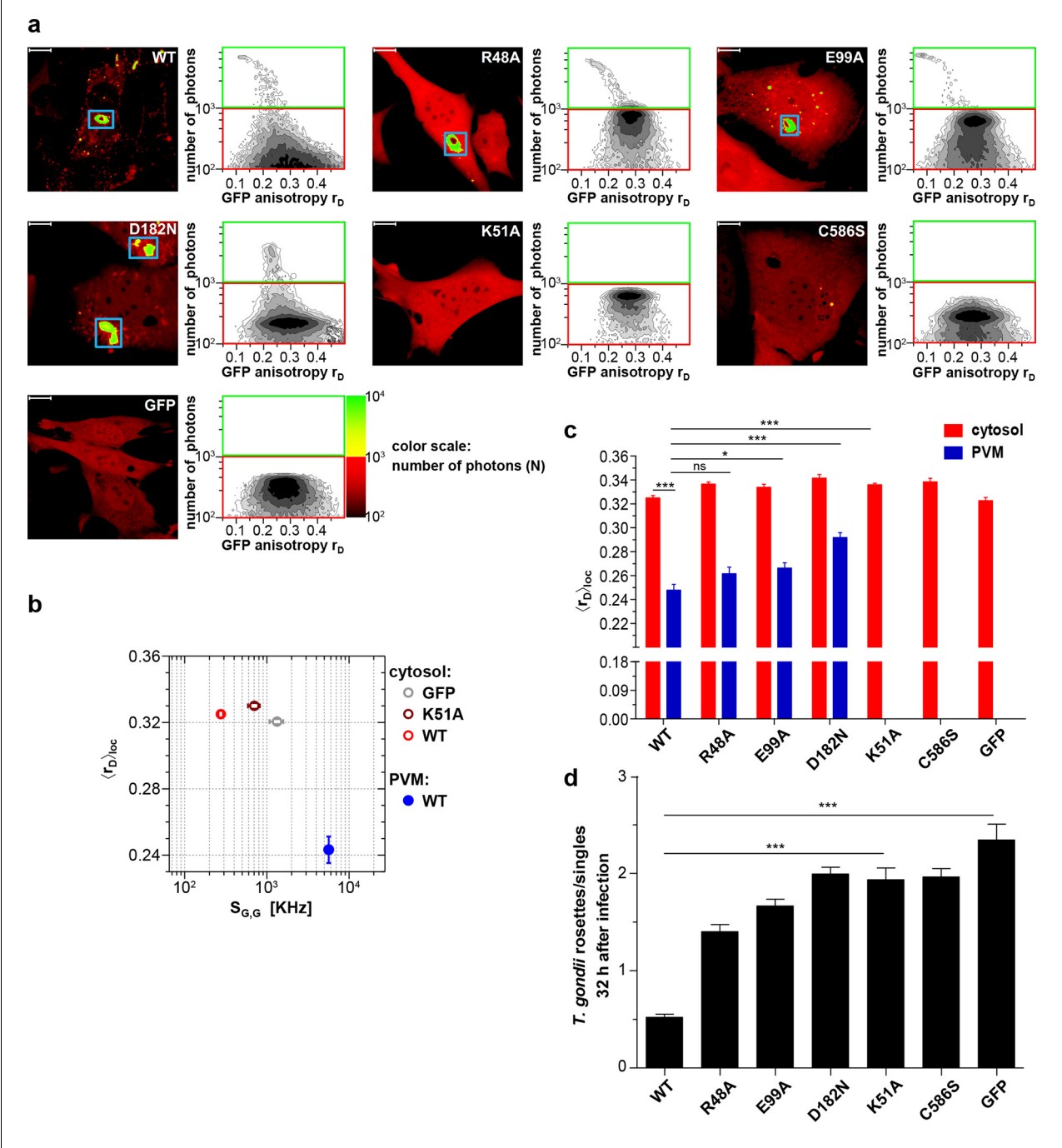

**Figure 2.** Intracellular homo-multimerization of WT and mutant mGBP2 at the PVM of *T. gondii* and parasite inhibition. Cells were pre-treated with IFNγ for 16 hr prior infection with *T. gondii* ME49 (**a**) Left panel. GFP fluorescence intensity images of G-mGBP2-WT, mutants MEFs or GFP highlighted with selections of pixels with low and high numbers of photons. Blue boxes mark the PVM area. Bars, 10 μm. Right panel. MFIS 2D-histograms of GFP $r_D$ on x axis vs. photon number per pixel on y axis. The pixels with low photon numbers (below 1000) are selected in red boxes and the pixels containing more than 1000 photons in green boxes. (**b**) Mean values of $<r_D>_{loc}$ and mean GFP $S_{G,G}$ were plotted for G-mGBP2-WT in the cytosol and at the PVM of *T. gondii* and for the K51A mutant and GFP in the cytosol. (**c**) Mean anisotropy $<r_D>_{loc}$ of WT and mutants in the cytosol and at the PVM (blue boxes in (**a**)). GFP expressing cells served as controls (ns=not significant; *p<0.05; **p<0.01; ***p<0.0001). (**d**) Replication inhibitory capacity of G-mGBP2-WT and mutants. After fixation *T. gondii* were stained with the α-SAG1 antibody and the cell nuclei with DAPI. Slides were analyzed by confocal microscopy. Replication inhibition was calculated by the ratio of *T. gondii* single parasites versus replicative units (rosettes) in at least 100 infected cells (***p<0.0001).

*Figure 2 continued on next page*

*Figure 2 continued*

The following figure supplement is available for figure 2:

**Figure supplement 1.** Spectroscopic characterization of G-mGBP2 WT in VLS in non-infected cells and at the PVM in *T. gondii* infected cells via homo-FRET assay.

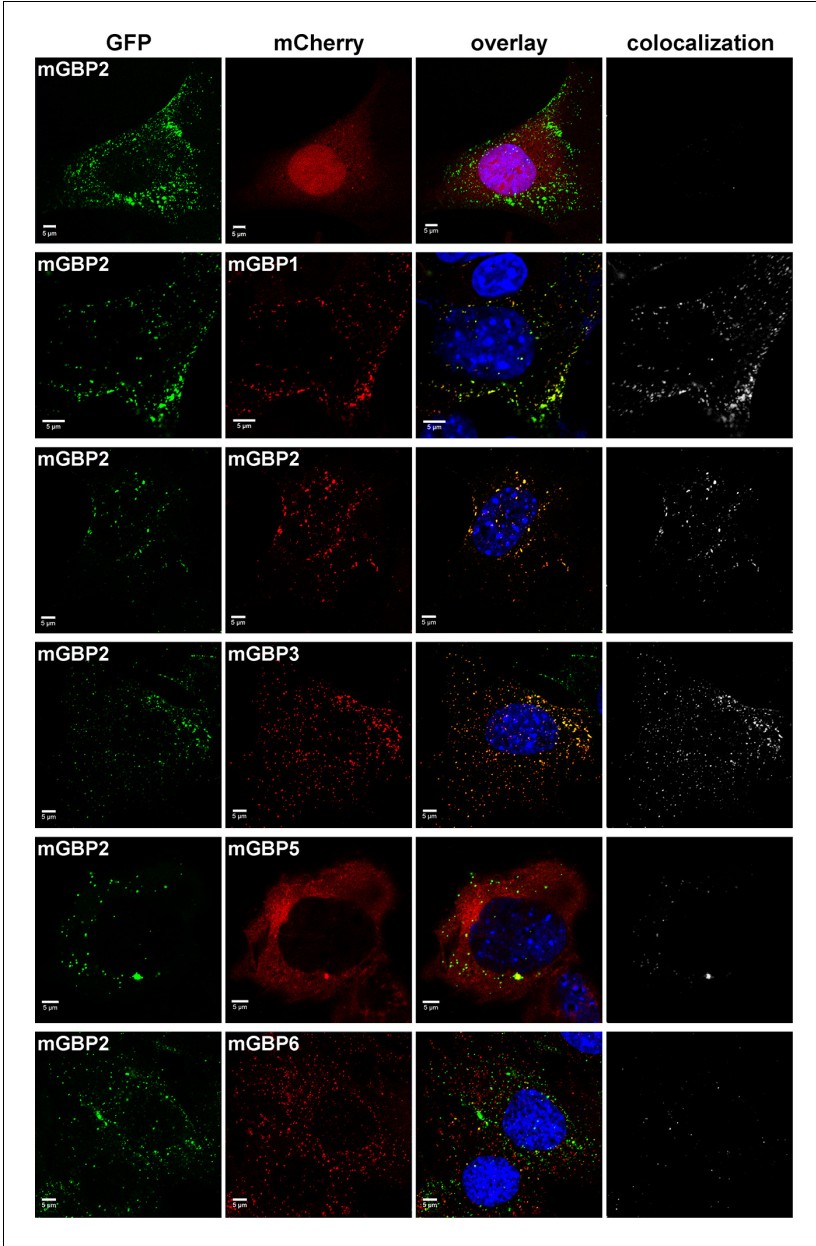

**Figure 3.** Intracellular colocalization of mGBP proteins. Subcellular localization of mGBPs was analyzed in G-mGBP2 coexpressing one of the mCh-mGBPs (1, 2, 3, 5 or 6). mCherry expressing cells served as controls. Cells were pre-treated with IFNγ for 16 hr. After fixation, nuclei were stained with DAPI. Glass slides were analyzed by confocal microscopy. Bars, 5 μm. Colocalization analysis was performed with Imaris (Bitplane).

The following figure supplement is available for figure 3:

**Figure supplement 1.** Expression analysis of coexpressed mGBP proteins.

displaying the frequency of pixels in color scales for the two localizations (red: cytosol, green: VLS), the VLS-population exhibits a correlated shift in the MFIS 2D-histogram of the FRET indicators $F_G/F_R$ and $<\tau_D>_f$ towards smaller values with respect to the population in the cytosol. This is a clear indicator for the presence of hetero-FRET, which proves the interaction between molecules. Furthermore, GFP $r_D$ was plotted versus $<\tau_D>_f$ as well as the G-mGBP2 concentration ($C_{G-mGBP2}$) derived from $F_G$ (see 'Determination of mGBP protein concentrations and binding curves', Materials and methods section) (*Figure 4b,c*, *Figure 4—figure supplement 1*). A $<\tau_D>_f - r_D$ diagram is essential to determine homo- and hetero-oligomerization between mGBPs sensed by hetero- and homo-FRET. *Figure 4b* illustrates the interpretation of a $<\tau_D>_f - r_D$ diagram based on the Perrin equation to visualize the effects on a donor-reference data set (green circle) by selective hetero- (red sphere) or homo-FRET (yellow sphere) or simultaneous homo- and hetero-FRET (orange sphere). Comparing G-mGBP2 MEFs (*Figure 4c*) with G-mGBP2/mCh-mGBP2 MEFs, both homo- and hetero-FRET were visible for the latter cells indicated by a simultaneous reduction of $<\tau_D>_f$ and an increase of $r_D$. Moreover, analyzing the cells individually, the anisotropy dropped with increasing G-mGBP2 concentrations. The variation of mGBP2 concentrations between individual cells allowed the estimation of the spatially resolved apparent dissociation constant ($K_{D,app}$) of the mGBP2 homomultimer of approx. 9 μM in the VLS (*Figure 4c*, upper right panel, black curve). Note that any interactions interfering with G-mGBP2 homomerization will result in a $K_{D,app}$-curve shifted upwards (purple curve).

To attain an overview of all experimental data, we computed the averaged values of $<\tau_D>_f$ and fluorescence intensity weighted anisotropy $<r_D>_{loc}$ for all cells of the specified FRET pair (*Figure 4c*, lower panels). Both in cytosol and in VLS, the strongest fluorescence lifetime reduction compared to the donor-only sample could be measured for combinations of G-mGBP2 with mCh-mGBP2 and to a lesser extent for mCh-mGBP1 and mCh-mGBP3 (*Figure 4a,c*), proving that mGBP1, 2, and 3 do not only colocalize but also directly interact. This could be confirmed by co-immunprecipitation (co-IP) experiments (*Figure 4—figure supplement 2*). Although no detectable lifetime reduction could be observed between G-mGBP2 and mCh-mGBP5, data showed a higher anisotropy compared to the donor reference, indicating interference of mGBP5 with mGBP2 homomerization (*Figure 4c*) Also, co-IP of mGBP2 and mGBP5 was observed (*Figure 4—figure supplement 2*), suggesting a differing mode of interaction between mGBP2 and mGBP5, which will be discussed below. No fluorescence lifetime reduction (*Figure 4c*, left panel), interaction-induced anisotropy increase (*Figure 4c*, right panel), or co-IP (*Figure 4—figure supplement 2*) could be observed for mGBP2 and mGBP6 coexpressing cells.

To elucidate the reason for the donor lifetime reduction in VLS by determining the fraction of FRET-active complexes ($x_{FRET}$) together with their FRET properties given by the rate constants of FRET ($k_{FRET}$), pixel-integrated MFIS-FRET analysis was applied by computing the FRET-induced donor-quenching decay $\varepsilon_{mix}(t)$ (*Equations 1–5*) to graphically display the FRET effect (*Figure 4d*). The larger drop of $\varepsilon_{mix}(t)$ (*Figure 4d*, upper panel) directly shows the difference in $x_{FRET}$ which proves that more interacting mGBP2 complexes reside in the VLS than in the cytosol. The FRET-induced donor decay $\varepsilon_{mix}(t)$ displays the interaction state of an ensemble of proteins, which includes both FRET-active and -inactive species. To separate the effects of both FRET-species on the decay, it is necessary to determine the characteristic $k_{FRET}$ of the populations in the cytoplasm and the VLS. The formally fitted decay curves (*Equations 1–5*) of FRET-active complexes $\varepsilon_{(D,A)}(t)$ are separately plotted (*Figure 4d*, lower panel), because this allows to remove the influence of the offset on the decay due to FRET-inactive species. The $\varepsilon_{(D,A)}(t)$ clearly differ for cytosol and VLS suggesting a higher degree of multimerization of mGBP2 in VLS. The $\varepsilon_{mix}(t)$-curve of a representative cell expressing G-mGBP2/mCh-mGBP6 (*Figure 4e*) had random fluctuations around 1, which is consistent with the data in *Figure 4a and c* showing no FRET events and confirms the absence of heteromeric complexes.

In summary, in the cytosol and VLS mGBP2 forms homo-multimers and hetero-multimers with mGBP1 and mGBP3, but not with mGBP6.

## Colocalization and hetero-FRET studies of mGBPs at the PVM of *T. gondii*

Individual members of the mGBP family are able to recruit to the PVM (*Degrandi et al., 2007*). To investigate the colocalization of several mGBPs at the PVM, G-mGBP2/mCh-mGBPx MEFs were

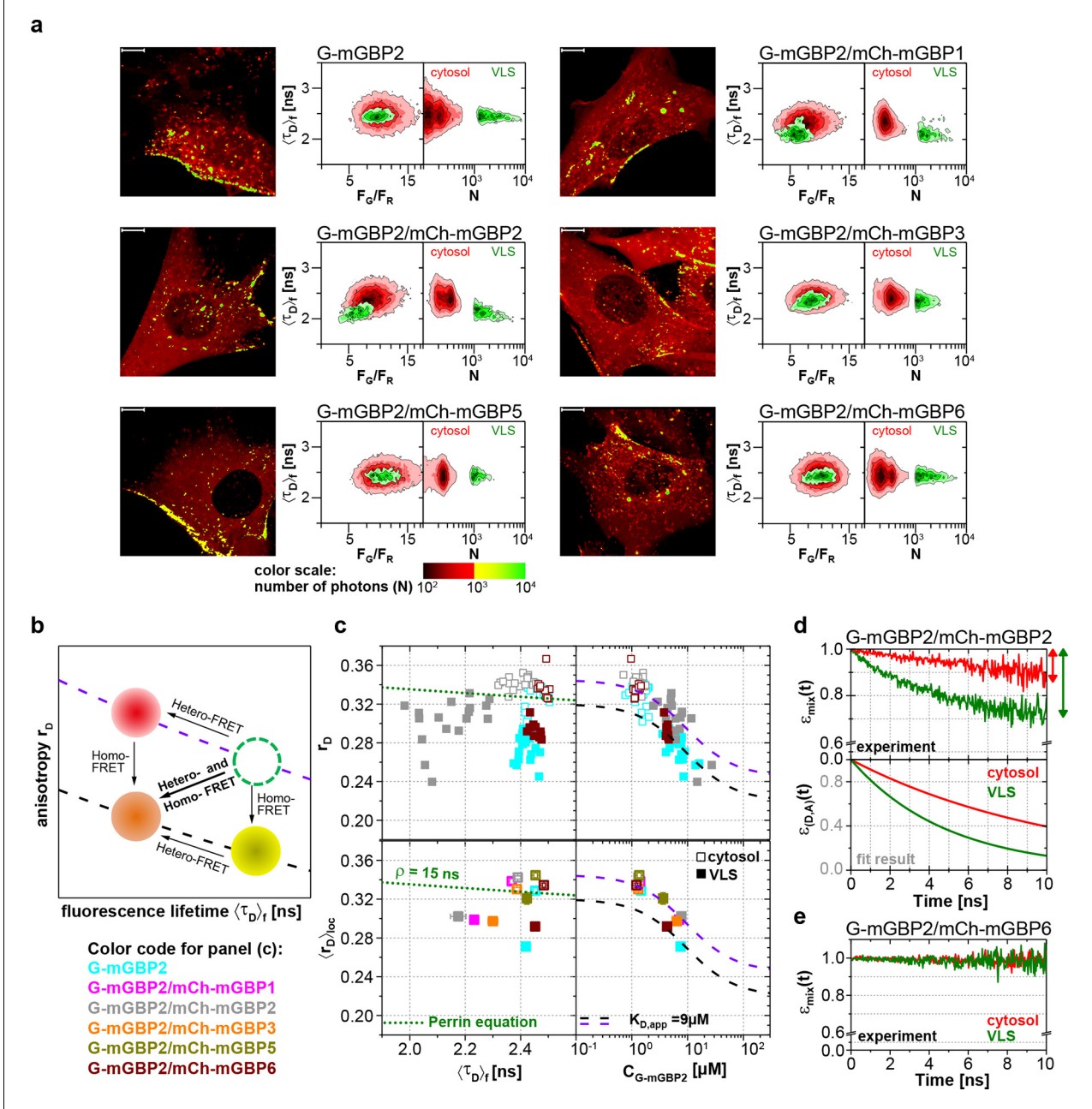

**Figure 4.** Intracellular homo- and hetero-multimerization of mGBPs. All cells were pre-treated with IFNγ for 16 hr prior investigation (**a**) Left panels. GFP fluorescence intensity images of G-mGBP2 or G-mGBP2/mCh-mGBP(1,2,3,5,6) MEFs highlighted with selections of pixels with different intensities. Bars, 10 µm. Right panels. Two MFIS 2D-histograms of GFP fluorescence lifetimes ($<\tau_D>_f$) on y axes, GFP/mCherry fluorescence intensity ratios ($F_G/F_R$) or photon number per pixel ($N$) on x axes. The pixel populations locating in cytosol ($N < 1000$: red island) and VLS ($N > 1000$: green island) were separated according to photon numbers. (**b**) Schematic 2D MFIS plot detailing the effects of hetero- and/or homo-FRET on a reference data set (green circle). The average GFP $<\tau_D>_f$ is plotted on the x axis from short to long, while the average steady-state $r_D$ is plotted on the y axis. For detailed explanation refer to results section. (**c**) Upper panel. For individual G-mGBP2, G-mGBP2/mCh-mGBP2or G-mGBP2/mCh-mGBP6 MEFs, mean values of $r_D$ in the cytosol (empty squares) and in the VLS (solid squares) were plotted against $<\tau_D>_f$ and G-mGBP2 concentrations ($C_{G-mGBP2}$). Lower panel. Mean anisotropy $<r_D>_{loc}$ values (average over all cells weighted by $C_{G-mGBP2}$) were plotted against $<\tau_D>_f$ or $C_{G-mGBP2}$. The two left panels contain an overlay calculated according to the Perrin equation: $r_D = r_0/(1 + \langle\tau_D\rangle_f/\rho_{global})$ with GFP fundamental anisotropy $r_0 = 0.38$ and rotational correlation time $\rho_{global}= 15$ ns. The two right panels are overlaid with function curves plotting $r_D = r_{max} - (r_{max} - r_{min}) \cdot C_{G-mGBP2}/(C_{G-mGBP2} + K_{D,app})$ which assumes a mGBP2 Langmuir binding model with an apparent dissociation constant $K_{D,app}$. In all donor-only experiments the formation of mGBP2 homo-multimers could be described by $K_{D,app} = 9$ µM, $r_{max} = 0.32$ and $r_{min} = 0.22$ (black curve). If other interaction processes interfere with homo-FRET between G-mGBP2 proteins, this curve is shifted upwards (violet curve) while keeping $K_{D,app}$ invariant ($r_{max} = 0.345$ and $r_{min} = 0.245$). (**d, e**) $\varepsilon_{mix}(t)$ and

*Figure 4 continued on next page*

Figure 4 continued

$\varepsilon_{(D,A)}(t)$ diagrams of a representative G-mGBP2/mCh-mGBP2 MEF (d) and G-mGBP2/mCh-mGBP6 MEF (e). The drop in $\varepsilon_{mix}(t)$ curves, as marked by the arrows, represents the species fractions of FRET-active complexes ($x_{FRET}$) in the VLS (green) and in the cytosol (red). In (d), the FRET rate constant ($k_{FRET}$) in the cytosol is 0.09 ns$^{-1}$ and in the VLS 0.20 ns$^{-1}$.

The following figure supplements are available for figure 4:

**Figure supplement 1.** Intracellular homo- and hetero-multimerization of mGBPs in cells.

**Figure supplement 2.** Immunoprecipitation analysis of mGBP proteins.

infected with *T. gondii*. (*Figure 5*). A colocalization of all investigated mGBPs with mGBP2 could be detected at distinct PVMs for each pairwise combination of proteins.

To investigate whether the colocalized mGBPs interact at the PVM, MFIS-FRET measurements were applied in G-mGBP2/mCh-mGBPx MEFs (*Figure 6*). A strong decrease of both FRET indicators, GFP fluorescence lifetimes $<\tau_D>_f$ and intensity ratio $F_G/F_R$, could be detected in the cytosol and at the PVM of G-mGBP2/mCh-mGBP1 and G-mGBP2/mCh-mGBP2 MEFs and, to a lesser extent, in G-mGBP2/mCh-mGBP3 MEFs (*Figure 6a,b*).

For individual cells, MFIS diagrams plotting the $r_D$ values against donor lifetimes $<\tau_D>_f$ and G-mGBP2 concentrations were generated (*Figure 6b* upper panels, *Figure 6—figure supplement 1*). The $K_{D,app}$-curves describing the relationship between $r_D$ and $C_{G-mGBP2}$ in uninfected cells (*Figure 4c*) fitted also very well to the infected situation (*Figure 6b*). The averaged values of $<\tau_D>_f$, $<r_D>_{loc}$ and $C_{G-mGBP2}$ over individual cells are depicted in *Figure 6b* (lower panels). An even stronger reduction in $<\tau_D>_f$ was observed at the PVM for combinations of G-mGBP2 with mCh-mGBP2 and to a lesser extent with mCh-mGBP1 and mCh-mGBP3 as compared to the VLS in uninfected cells (*Figure 4c*), proving that the observed colocalization at the PVM (*Figure 5*) enables direct protein interactions. For G-mGBP2/mCh-mGBP5 MEFs the situation is more complex: in the cytosol the anisotropy was slightly increased but the donor lifetime was unchanged, whereas at the PVM an increase in anisotropy was absent (*Figure 6b*, lower right panel). In G-mGBP2/mCh-mGBP6 MEFs no interactions were detected, neither in the cytosol nor at the PVM.

The FRET-related donor quenching $\varepsilon_{mix}(t)$ of one representative G-mGBP2/mCh-mGBP2 cell (*Figure 6c*) exhibited a larger drop, which indicates a higher $x_{FRET}$, i.e. more interacting protein complexes were located at the PVM compared to VLS in uninfected cells (*Figure 4d*). Nevertheless, their slopes ($k_{FRET}$) of $\varepsilon_{(D,A)}(t)$ are comparable within the precision of the analysis (*Figure 6c*, green dashed line), suggesting an unchanged local environment in the oligomer. Furthermore, the $\varepsilon_{mix}(t)$ diagram for one representative G-mGBP2/mCh-mGBP6 cell revealed no interaction between these mGBPs.

In conclusion, mGBP2, besides its homo-interaction, directly interacts with mGBP1 and, to a lesser extent, with mGBP3 at the PVM. Although other mGBPs, such as mGBP5 and mGBP6 were recruited to the same PVMs, no direct interaction could be detected suggesting the formation of specific mGBP supramolecular complexes.

## Quantitative species-resolved pixel-integrated MFIS-FRET analysis of mGBPs multimers

In addition to the formal analysis by *Equations 1–5* (*Figure 6—figure supplement 2*) of the hetero-FRET data, an additional inspection of the time-resolved donor anisotropy ($r_D(t)$) (*Figure 7a*) revealed that cells with a higher mGBP2 concentration ($C_{mGBP2}$) exhibited a larger drop in initial anisotropy, which is evidence for ultrafast depolarization processes due to the formation of densely packed mGBP2 homo-oligomers with multiple GFPs. These processes were too fast to be resolved by hetero-FRET analysis (*Figure 6c*), but combining both homo- and hetero-FRET, global pattern based, pixel-integrated MFIS-FRET analysis could be performed to resolve the individual mGBP species (*Figure 7b and c*) and to characterize the composition of FRET-active homo- and hetero-complexes of mGBP2 (*Equation 6, 7*) for the distinct localizations. The information content in the experimental fluorescence decays is restricted by their noise (*Kollner and Wolfrum, 1992*). Given the limited amount of photons of the pixel-integrated fluorescence intensity histograms, the pattern fit uses structural information of molecular simulations (*Figure 7—figure supplement 1*) to obtain

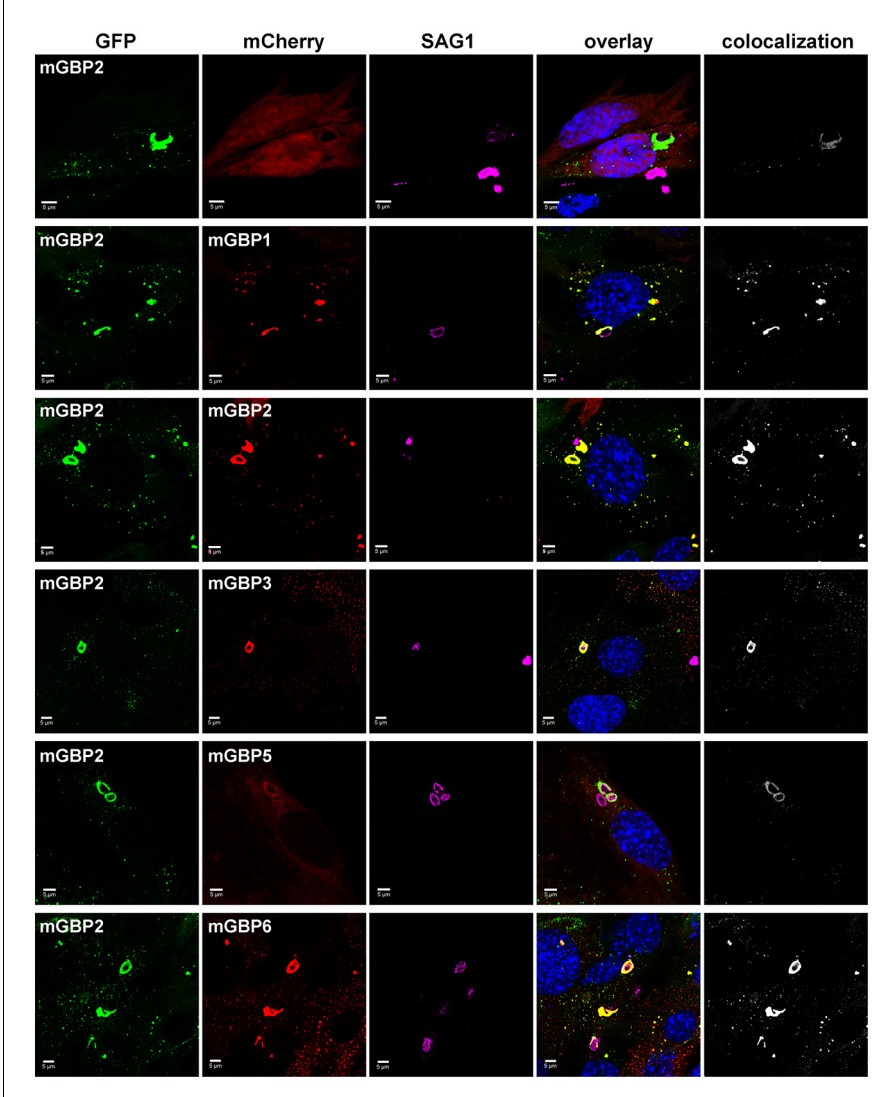

**Figure 5.** Intracellular colocalization at the PVM of *T. gondii* and enrichment of mGBP proteins. Recruitment and colocalization of mGBPs was analyzed in G-mGBP2/mCh-mGBP(1,2,3,5,6) MEFs. mCherry expressing cells served as controls. Cells were stimulated with IFNγ for 16 hr and subsequently infected with *T. gondii* for 2 hr. After fixation, *T. gondii* were stained with an α-SAG1 antibody and cell and *T. gondii* nuclei with DAPI. Glass slides were analyzed by confocal microscopy. Bars, 5 μm. Colocalization analysis was performed with Imaris (Bitplane).

population fractions of all species. The structural information is based on prior knowledge of the dimerization interface (*Vopel et al., 2014*) and on Monte Carlo simulations of the linkers connecting the fluorescent proteins to the GBPs (see 'Monte Carlo sampling of the donor-acceptor conformational space of mGBP2 dimer', Materials and methods section) (*Evers et al., 2006*; *Pham et al., 2007*). The obtained species fractions of mGBP2 monomers, homo- or hetero-dimers and oligomers are displayed in *Figure 7c*. The homo- and hetero-dimer formation is very similar in G-mGBP2 MEFs and G-mGBP2/mCh-mGBP1, 2 or 3 MEFs as expected for the highly conserved GTPase-domains of mGBPs. Dimeric complexes are primarily formed with a small fraction of monomers in the cytosol (*Figure 7c*, middle panel, see methods, *Equation 13*). The obtained $K_{D,dim}$ of ~24 nM is close to previous biochemical studies (*Kravets et al., 2012*). In the VLS an equilibrium of mGBP dimers and oligomers existed which was shifted towards oligomers with increasing protein concentration so that, the fraction of oligomers at the PVM is even higher than in the VLS. However, the dissociation constants for oligomerization $K_{D,oligo}$ differ significantly between the mGBPs: 70 μM for G-mGBP2/

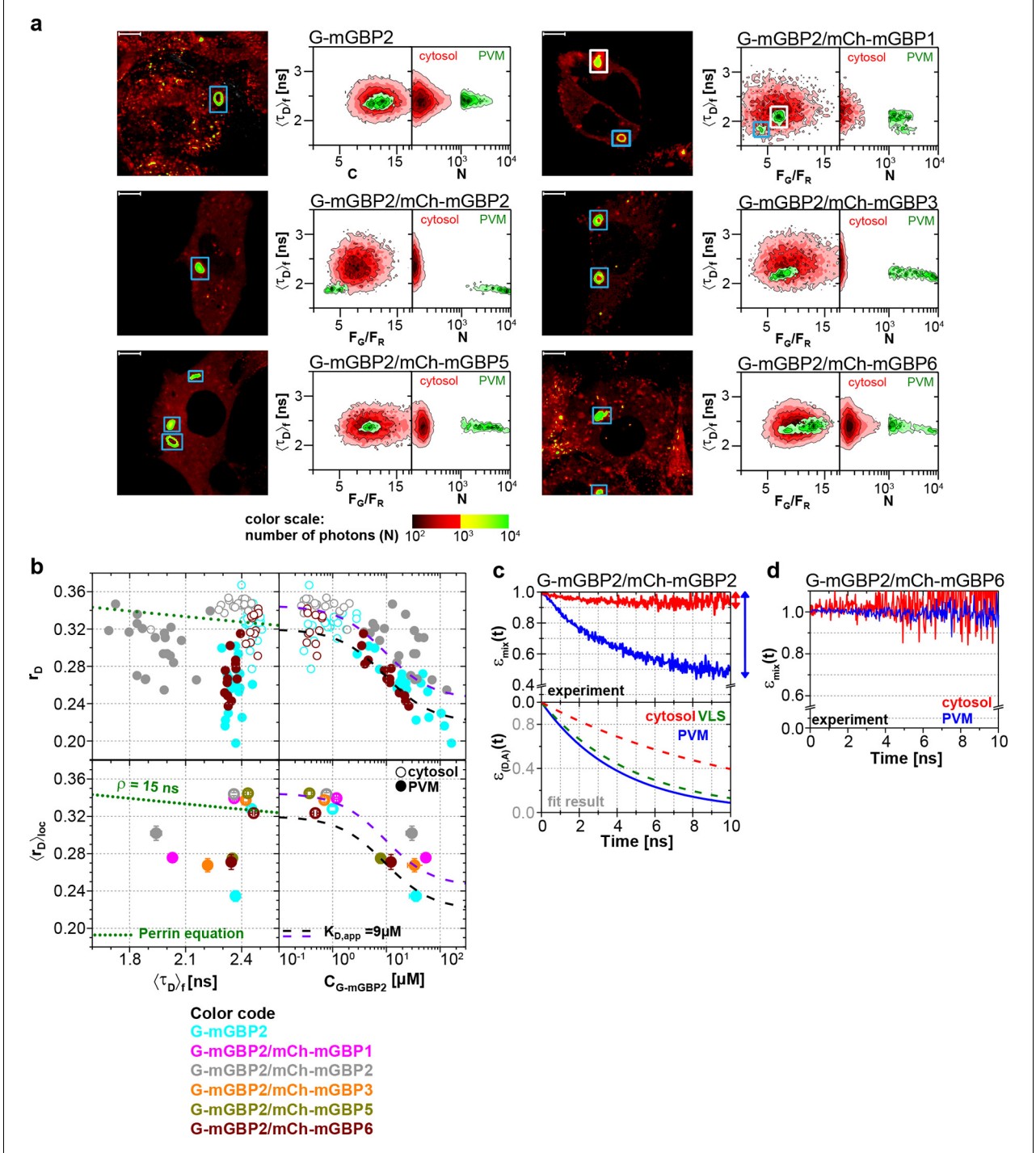

**Figure 6.** Intracellular homo- and hetero-multimerization of mGBPs at the PVM of *T. gondii*. All cells were pre-treated with IFNγ for 16 hr prior investigation (**a**) Left panels. GFP fluorescence intensity images of living G-mGBP2 or G-mGBP2/mCh-mGBP(1,2,3,5,6) MEFs infected with *T. gondii* highlighted with selections of pixels within different intracellular localizations. Right panels. Two MFIS 2D-histograms of GFP $\langle\tau_D\rangle_f$ on y axes, GFP/mCherry $F_G/F_R$ and photon number per pixel (*N*) on x axes. The pixel populations locating in cytosol (*N* < 1000: red island) and at the PVM (*N* > 1000: green island) were separated according to photon numbers. (**b**) Upper panel. For individual G-mGBP2, G-mGBP2/mCh-mGBP2 or G-mGBP2/mCh-mGBP6 MEFs pixel averages of $r_D$ in the cytosol and at the PVM were plotted against $\langle\tau_D\rangle_f$ or $C_{\text{G-mGBP2}}$. Lower panel. Averages of $\langle r_D\rangle_{\text{loc}}$ were plotted against $\langle\tau_D\rangle_f$ and $C_{\text{G-mGBP2}}$. Please refer to *Figure 4c* for further information on the legend and overlaid curves. (c, d) $\varepsilon_{mix}(t)$ and $\varepsilon_{(D,A)}(t)$ diagrams of a representative *T. gondii* infected G-mGBP2/mCh-mGBP2 MEF (**c**) and G-mGBP2/mCh-mGBP6 MEF (**d**). The drop in $\varepsilon_{mix}(t)$ curves, as marked by the arrows, represents $x_{FRET}$ at the PVM (blue) and in the cytosol (red). The dashed curves representing the $\varepsilon_{(D,A)}(t)$ diagrams of G-mGBP2/

*Figure 6 continued on next page*

*Figure 6 continued*

mCh-mGBP2 interactions in the cytosol (red) and VLS (green) in uninfected cells are inserted for comparison from *Figure 4d*. In (c), $k_{FRET}$ at the PVM is 0.24 ns$^{-1}$.

The following figure supplements are available for figure 6:

**Figure supplement 1.** Intracellular homo- and hetero-multimerization of mGBPs in *T. gondii* infected cells.

**Figure supplement 2.** Quantitative MFIS-FRET analysis of mGBP2 hetero-multimerization in living IFNγ stimulated cells.

mCh-mGBP1, 8 μM for G-mGBP2/mCh-mGBP2 and 208 μM G-mGBP2/mCh-mGBP3 (*Figure 7c*, lower panel).

Global analysis of G-mGBP2 MEFs and G-mGBP2/mCh-mGBP2 MEFs, revealed the heterogeneity in size of the mGBP2 oligomers via the broad distribution of FRET rate constants for small and large oligomers, $k_{Olig,s}$ and $k_{Olig,l}$, respectively (*Figure 7d*). While $k_{Olig,s}$ did not change with increasing protein concentration, $k_{Olig,l}$ increased and reached a saturation level of ~15 ns$^{-1}$ at ~50 μM (*Figure 7d*, red line), which is expected for a maximal local packing of FRET acceptors around the donor (see 'Maximum FRET rate constants', Materials and methods section) and proved the growth of oligomers. Notably FRET senses only the local environment in a distance range limited to ~10 nm, however the continuous increase in brightness suggests also the formation of larger oligomers. Therefore we introduced scanning fluorescence intensity distribution analysis (FIDA) ((*Kask et al., 2000*), 'Scanning fluorescence intensity distribution analysis (FIDA) for determination of oligomer size', Materials and methods section) to determine the mean number and brightness of the large oligomers for all pixels of the PVM in one infected MEF. The obtained oligomer brightness allowed us to derive the mean number of mGBP2 units in an oligomer using the specific brightness of one GFP under these measurement conditions. With increasing local mGBP2 concentration, scanning FIDA suggests also an increasing oligomer size (*Figure 7e*). The mean number of mGBP2 monomer units in the oligomer ranges between 1000 and 6000 at the PVM. Remarkably the FRET rate constants in large oligomers $k_{Olig,s}$ saturated at approximated 2000 monomer units, which corresponds to a total local concentration of mGBP2 monomer units of ~ 30 μM (*Figure 7e*).

In summary, with increasing protein concentration the fraction of mGBP2 dimers decreases due to the formation of large oligomers of heterogeneous size. The formation of mGBP2 homo-oligomers is preferred over heteromers with mGBP1 and mGBP3 as $K_{D,oligo}$ dropped by a factor of 9 and 25, respectively. The mean size of large mGBP2 oligomers can reach up to several thousand monomer units.

## mGBP2 directly targets the parasite membrane

mGBP2 was shown to rapidly accumulate at the PVM after active invasion of the parasite in IFNγ activated cells (*Degrandi et al., 2013*). To further investigate the spatio-temporal behavior of mGBP2, 3D live cell imaging was performed in mGBP2$^{-/-}$ MEFs stably expressing G-mGBP2 or mCh-mGBP2 (*Figure 8* and *Videos 1–3*). mGBP2 localized in VLS of heterogeneous size, morphology, and velocity within the cytosol. In IFNγ stimulated uninfected cells the diameter of VLS reaches up to several microns. No obvious directional movement could be observed (*Video 1*). After *T. gondii* infection of IFNγ stimulated MEFs, mGBP2 accumulated rapidly at the PVM (*Figure 8a,b* and *Video 1*). Image analysis revealed that accumulation initiated simultaneously at different sites around the PVM (*Figure 8b*). Quantification of the overall G-mGBP2 fluorescence in regions containing the PVM and the remaining cell revealed a constant reduction of the cytosolic and VLS G-mGBP2 concentrations after infection, paired with a reciprocal increase at the PVM (*Figure 8c*). Thus, accumulation of mGBP2 at the PVM occurs by redistribution of the protein, leading to a depletion of mGBP2 reservoirs and a reduction of the number of VLS (*Figure 8d*) within the cytosol. However, no directional movement of VLS towards the parasite could be observed (*Video 1*).

After accumulation of mGBP2 at the PVM of *T. gondii*, different fates of the parasite could be observed within the recording period by live cell imaging. mGBP2 remained at the PVM for more than 16 hr without any noticeable change in PVM or parasite morphology (not shown), mGBP2 penetrated through the PVM into the vacuolar space and accumulated at the parasite membrane

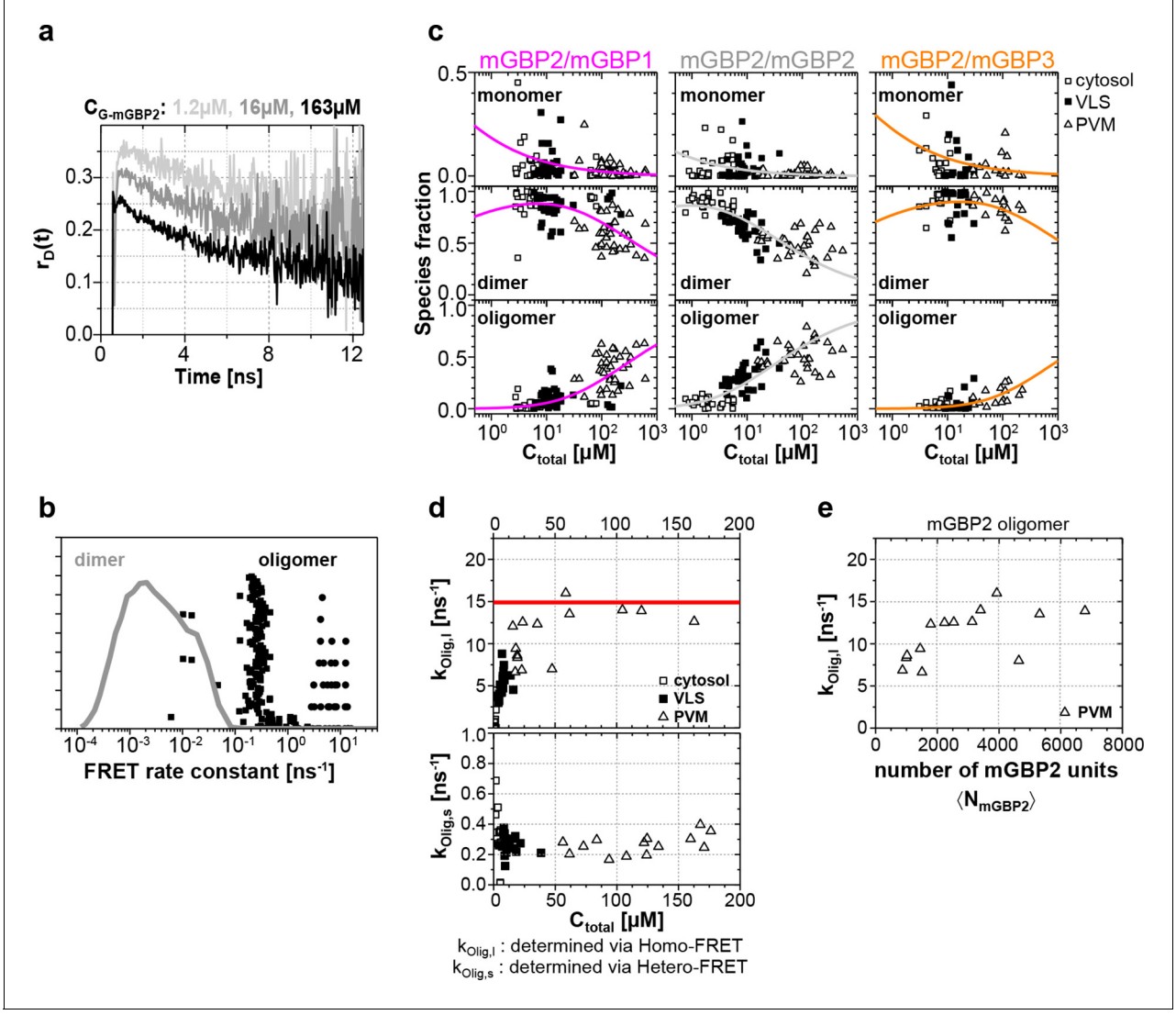

**Figure 7.** Species-resolved analysis of mGBP2 homo- and hetero-complexes. (a) G-mGBP2 MEFs with higher concentration exhibited larger quasi instantaneous drop of $r_D(t)$ from its initial value of ~0.35, which proves the appearance of a very fast depolarization process due to homo-FRET in mGBP2 oligomers. (b) Distribution of FRET rate constants ($k_{FRET}$) for mGBP2 dimer (gray curve) and oligomer species (black symbols). Small (black squares) and large (black dots) oligomers, as formally differentiated in the pattern-based MFIS-FRET analysis, show generally higher $k_{FRET}$ than that of the mGBP2 dimer estimated by the MC simulation. (c) Concentration dependence of the three mGBP species (monomer, dimer and oligomer) obtained by the global pattern fit (*Equations 6 and 7*) of $r_{mix}(t)$ and $\varepsilon_{mix}(t)$ for two localizations VLS and PVM. The line depicts the fit ('Pattern based pixel-integrated MFIS-FRET analysis' and 'Determination of dissociation constants', Materials and methods section) to the corresponding binding equilibrium with $K_{D,dim}$, and $K_{D,app-oligo}$ (values are given in the main text). (d) Concentration dependence of FRET rate constants for mGBP2 oligomers which formally differentiated as small ($k_{Olig,s}$) and large ($k_{Olig,l}$). (e) $k_{Olig,l}$ versus the number of monomer units in mGBP2 multimers at the PVM determined by scanning FIDA (see 'Scanning fluorescence intensity distribution analysis (FIDA) for determination of oligomer size', Materials and methods section).

The following figure supplement is available for figure 7:

**Figure supplement 1.** Sample mGBP2 dimer conformations by MC molecular simulation.

(*Figure 8e* and *Video 2*), or the mGBP2-associated PVM acquired a rounded shape immediately followed by disruption of the PVM and subsequent accumulation of mGBP2 at the parasite membrane (*Figure 8f* and *Video 3*). Importantly, the behavior of mGBP2 was independent of the mCherry or GFP fusion.

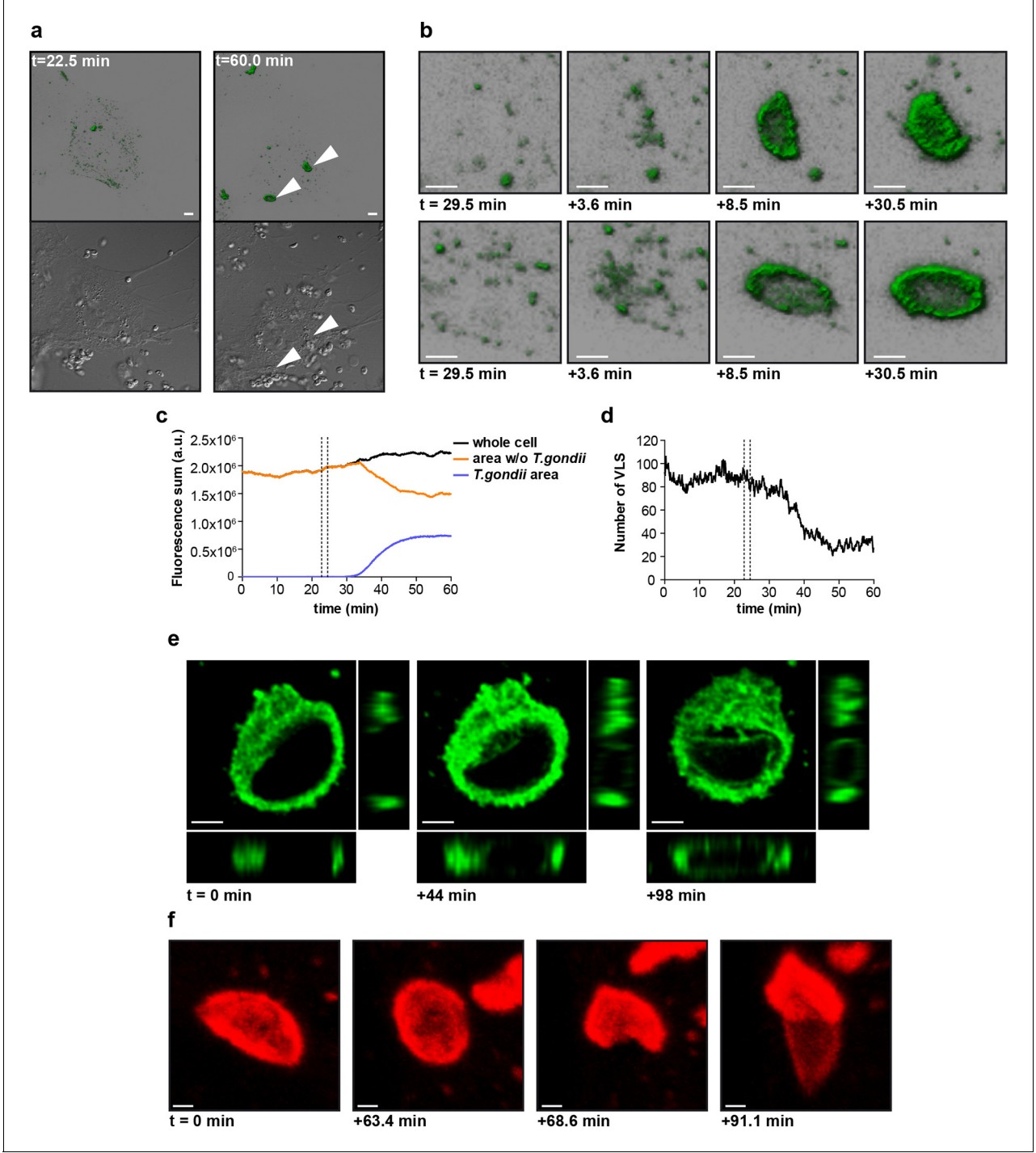

**Figure 8.** Live-cell imaging of mGBP2 in *T. gondii* infection. (a) G-mGBP2 MEFs were treated o/n with IFNγ and infected with *T. gondii* ME49. Living cells were observed by confocal microscopy at 37°C and a z-stack was recorded every 5–10 s. 4D data were processed and rendered in normal shading mode (upper panels) and the DIC images are displayed (lower panels) for the indicated time points. One out of at least 3 similar experiments is shown. Bar = 5 μm. (b) Magnification from *Video 1* and *Figure 8a* of G-mGBP2 accumulation around two *T. gondii* parasites at time points indicated. Bar = 2 μm. (c) Quantification of the total fluorescence intensity over the indicated voxels from *Video 1*. Vertical lines indicate the time points of *T. gondii* infection of MEFs. One representative analysis out of at least 3 similar experiments is shown. (d) Number of cytosolic VLS with at least approx. 0.25 μm diameter from *Video 1* over time. Fluorescence signals close to the *T. gondii* area were excluded from the analysis. Vertical lines indicate the time points of *T. gondii* infection of MEFs. One representative analysis out of at least 3 similar experiments is shown (e) XY, XZ, and YZ projections of G-mGBP2 around one *T. gondii* PVM are shown for the indicated time points. Bar = 2 μm. (f) Maximum intensity projections of mCh-mGBP2 around one *T. gondii* are shown for the indicated time points. Bar = 1 μm.

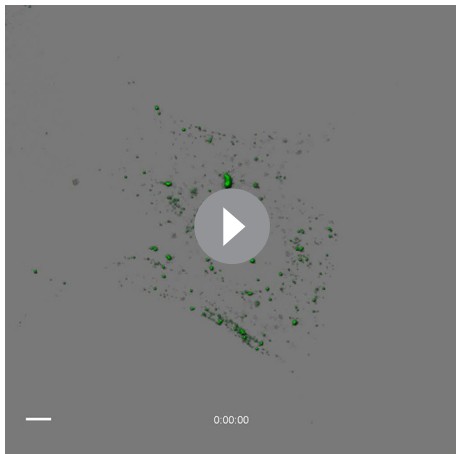

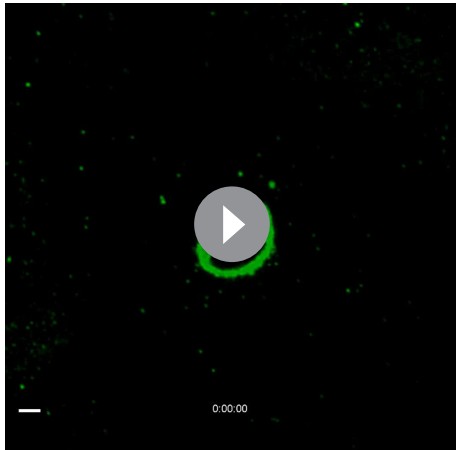

**Video 1.** mGBP2$^{-/-}$ MEFs transduced with G-mGBP2 were treated o/n with IFNγ and infected with *T. gondii*. The living cells were observed with a confocal microscope at 37°C and a z-stack was recorded every 5–10 s. 4D data were processed and rendered in normal shading mode. Bar = 5 µm.

**Video 2.** mGBP2$^{-/-}$ MEFs transduced with G-mGBP2 were treated o/n with IFNγ and infected with *T. gondii*. The living cells were observed with a confocal microscope at 37°C and a z-stack was recorded every 5–10 s. 4D data were processed and rendered as maximum intensity projection. Bar = 2 µm.

Additionally, the events following mGBP2 recruitment to the PVM were documented and quantified. For this, IFNγ stimulated G-mGBP2 MEFs were infected with *T. gondii* for 6 hr, fixed and the plasma membrane of *T. gondii* was stained with anti-SAG1. To determine the precise localization of mGBP2 at this time point, intensity profiles of G-mGBP2 and Alexa633-SAG1 were determined encompassing the PVM, the plasma membrane of the parasite and the cytosol of the parasite (*Figure 9*). A total of 110 intracellular mGBP2-positive *T. gondii* PVs out of two independent experiments were evaluated. About 1.8% of the parasites acquire mGBP2 on the plasma membrane without apparent loss of PV integrity (*Figure 9c*). For 37.1% of counted parasites disruption of PVM and direct targeting of mGBP2 to the plasma membrane of the parasite was observed (*Figure 9b*). The remaining 61.1% revealed mGBP2 targeting at the PVM without apparent disruption or permeabilization and targeting of the parasite plasma membrane (*Figure 9a*). Occasionally, after 6 hr of infection, parasites with very aberrant SAG1 localization were observed, providing evidence that these parasites were already non-viable. In such cases G-mGBP2 fluorescence inside the cytosol of the parasite could be found, suggesting a loss of the membrane integrity of the parasite (*Figure 9d*).

As previously reported, a rapid colocalization of mGBP2 with the PV of *T. gondii* type II strain ME49 but not of *T. gondii* type I strain BK in IFN-γ–activated MEFs was observed (*Degrandi et al., 2007*). After infection with *T. gondii* ME49, selective permeabilization experiments revealed that immunofluorescence labeling of SAG1 at the *T. gondii* plasma membrane could be detected for mGBP2-positive PVMs in

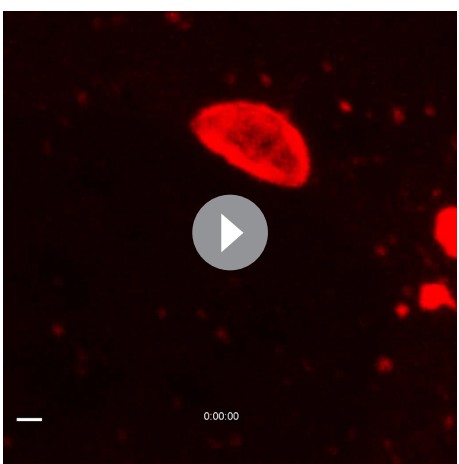

**Video 3.** mGBP2$^{-/-}$ MEFs transduced with mCh-mGBP2 were treated o/n with IFNγ and infected with *T. gondii*. The living cells were observed with a confocal microscope at 37°C and a z-stack was recorded every 5–10 s. 4D data were processed and rendered as maximum intensity projection. Bar = 1 µm.

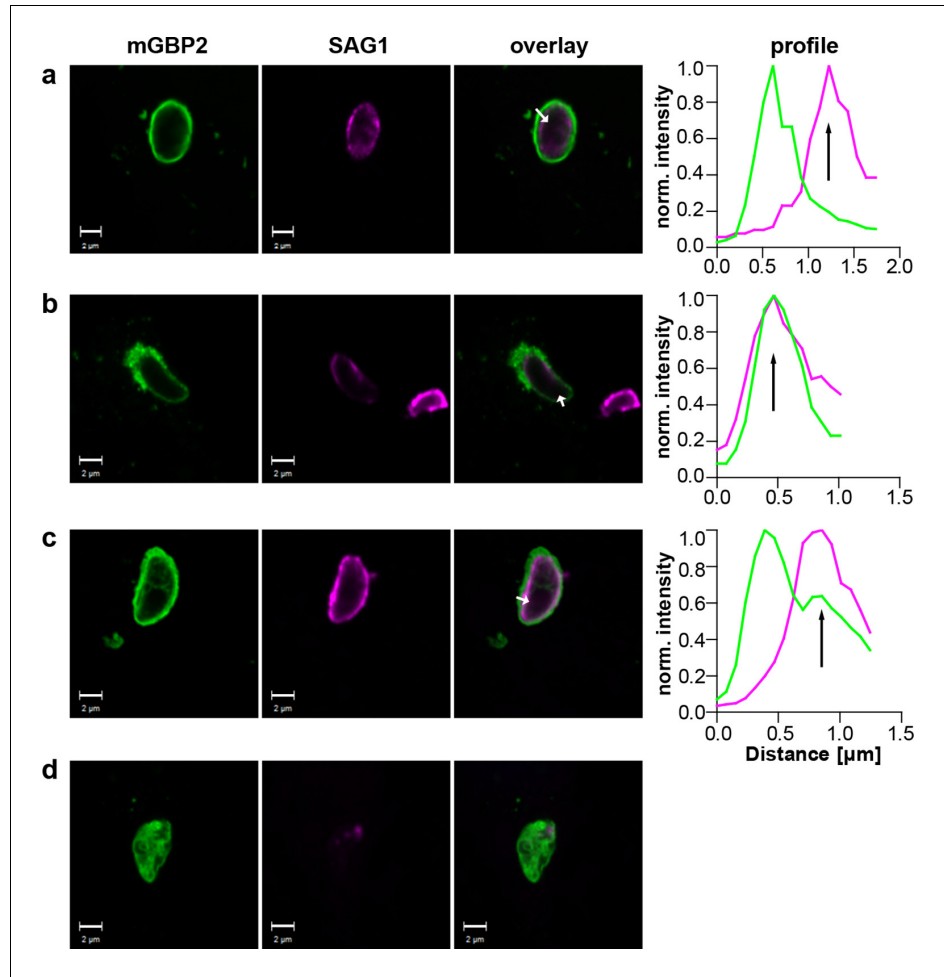

**Figure 9.** Localization of mGBP2 at the PVM, the plasma membrane, or the cytosol of *T. gondii*. G-mGBP2 cells were stimulated with IFNγ for 16 hr and subsequently infected with *T. gondii* ME49 for 6 hr. After fixation, *T. gondii* were stained with an α-SAG1 antibody. Glass slides were analyzed by confocal microscopy. Bars, 2 μm. Profiles show individually normalized intensities of GFP (mGBP2, green) or Alexa633 (SAG1, magenta) fluorescence along the indicated white arrows. Black arrows indicate the localization of the *T. gondii* plasma membrane, as identified by the SAG1 staining. (a) Example of mGBP2 accumulation at the PVM of *T. gondii* without disruption or permeabilization of the PVM. (b) Example of mGBP2 accumulation at the plasma membrane of *T. gondii* with obvious disruption of the PVM. (c) Example of mGBP2 accumulation at the plasma membrane of *T. gondii* without apparent PVM disruption. (d) Example of *T. gondii* death and accumulation of mGBP2 in the cytosol of the parasite.

**Table 2.** G-mGBP2 cells were stimulated with IFNγ for 16 hr and infected with *T. gondii* ME49 or BK strains for 2 hr. After fixation and permeabilization with the indicated amounts of saponin, *T. gondii* were stained with an α-SAG1 antibody and DAPI. *T. gondii* were counted and categorized according the indicated mGBP2 and SAG1 fluorescence. N.d = not detected.

|  | ME49 *T. gondii* | | | BK *T. gondii* | | |
|---|---|---|---|---|---|---|
|  | mGBP2+ SAG1- | mGBP2+ SAG1+ | mGBP2- SAG1+ | mGBP2+ SAG1- | mGBP2+ SAG1+ | mGBP2- SAG1+ |
| w/o Saponin | 50% | 38% | 12% | n.d. | n.d. | 3% |
| 0,15% Saponin | n.d. | 57% | 43% | n.d. | 1% | 99% |

the absence of saponin. In contrast, after infection with the virulent BK *T. gondii*, almost no SAG1-labeled parasites could be detected (*Table 2*). Please note that after saponin permeabilization virtually all ME49 or BK parasites could be labeled with anti-SAG1. This shows that targeting of mGBP2 to the PVM promotes permeabilization or disruption of the PVM, allowing influx of proteins into the PV space.

Additionally, we have monitored the influx of cytosolic mCherry protein into the PV space after PVM disruption of GFP-mGBP2 positive *T. gondii* PV (*Video 4*). This observation corroborates former experimental approaches, showing a disruption of the PVM after IRG recruitment (*Zhao et al., 2009b*).

Taken together, these observations show direct evidence that mGBP2 promotes PVM permeabilization and disruption and provide novel evidence that mGBP2 translocates into the PV space targeting the plasma membrane of the parasite, presumably delivering a direct attack on the parasite.

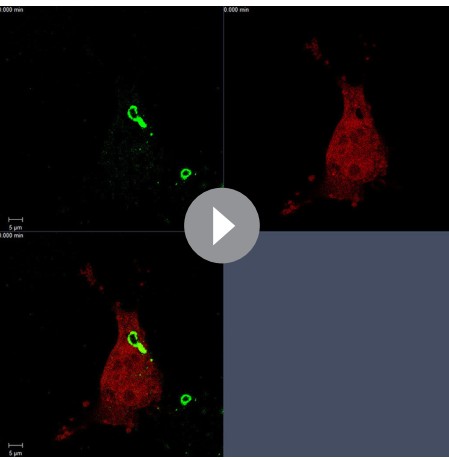

**Video 4.** mGBP2⁻/⁻ MEFs transduced with G-mGBP2 and cytosolic mCherry were treated o/n with IFNγ and infected with *T. gondii*. The living cells were observed with a confocal microscope at 37°C.

## Discussion

The localization, molecular dynamics, interactions, and the formation of mGBP supramolecular complexes in the context of defense against *T. gondii* could be directly visualized in living cells using MFIS and live cell imaging within this study. Our data demonstrate that GTP binding and hydrolysis as well as membrane anchoring enable the pre-assembly of multimeric complexes containing mGBP2 in VLS. mGBP2/mGBP2, mGBP2/mGBP1 and mGBP2/mGBP3 complexes in the form of dimers and multimers with distinct composition are recruited at considerably high concentrations (10–200 µM) to the PVM of *T. gondii*. Moreover, the GTPase activity and isoprenylation of mGBP2 are crucial for the control of *T. gondii* proliferation within the PV. Eventually, mGBP2 multimers target the plasma membrane of *T. gondii*, thus establishing the immune function of GBPs to directly attacking intracellular pathogens.

To extract structural information from the MFIS-FRET data (*Kalinin et al., 2012*), we performed Monte Carlo sampling of the donor-acceptor conformational space of the mGBP2 dimer to compute the expected FRET parameters ('Monte Carlo sampling of the donor-acceptor conformational space of mGBP2 dimer', Materials and methods section, *Figure 7—figure supplement 1b*). The sterically accessible volume of flexibly attached fluorescent proteins (green (GFP) and red (mCherry)) are depicted as fuzzy clouds. The prediction that more than 60% of all D-A configurations are FRET-inactive due to their large distances between the fluorophores is confirmed by the formal MFIS-FRET analysis (*Figure 6—figure supplement 2d*). Our data argue that GTP binding is a prerequisite to induce dimer-and multimerization of mGBP2 in living cells. Indeed, the simulated FRET parameters of the mGBP2 homodimer (*Figure 7—figure supplement 1b–d*) interacting via the GTPase domains are in good agreement with MFIS pixel integrated analysis (*Figure 4d*, *6c*, *Figure 6—figure supplement 2*). Moreover, the K51A mutant, which is predicted to be predominantly nucleotide-free (*Kravets et al., 2012*), shows higher anisotropy values compared to WT, is entirely delocalized in the cytosol, and is monomeric in living cells (this study). However, GTPase-domain dimerization is not sufficient to determine the targeting of mGBP2 to the PVM.

Interestingly, individual murine and human GBPs (hGBPs) harbor C-terminal CaaX motifs (GBP1, GBP2, GBP5), targeting them for isoprenylation, which provides anchorage to different membranous compartments distributed within the host cell (*Degrandi et al., 2007*; *Britzen-Laurent et al., 2010*; *Vestal et al., 2000*). As described for hGBP1, the dimerization of the GTPase-domains enables contact formation between the two C-terminal α13 helices resulting in a juxtaposition which is crucial for

their membrane localization through the attached farnesyl groups (*Vopel et al., 2014*). The purified CaaX mutant of mGBP2 (C586S) shows GTP binding and hydrolysis properties as well as nucleotide dependent dimerization like the WT protein (*Figure 1—figure supplement 1*). However, the C586S mGBP2 mutant renders the protein non-functional and it is found ubiquitously within the cytosol. Noteworthy, the isoprenylation mutant C586S shows similar localization and anisotropy values as the K51A mutant in living cells, also indicating a monomeric species. Altogether these studies suggest an assembly mechanism for mGBP2 complexes in living cells that connects the GTPase activity of mGBP2 with membrane association leading to the stabilization of mGBP2 multimers, which is essential for its biological function. Moreover, MFIS measurements with high-precision FRET and brightness analysis allowed us to characterize the dynamic equilibrium between mGBP2 multimers. Their size distribution is heterogeneous ranging from dimers to large multimers (*Figure 7c and d*). The dependence of FRET rate constants on the mGBP2 concentration and their saturation level proves dense packing of the mGBP2 protomers in multimers ('Maximum FRET rate constants', Materials and methods section) as suggested for the related mechanochemical GTPase dynamin forming large helical oligomers (*Faelber et al., 2011*). While FRET characterizes the molecular environment of GFPs, scanning FIDA shows that the average number of mGBP2 units in the oligomers can reach several thousands. Considering the predicted size of the mGBP2 monomer (~ $4 \times 6 \times 12$ nm, according to PDB-ID 1F5N of hGBP1), it is expected that the oligomers should reach a size of several hundred nanometers. Remarkably, confocal live cell imaging (*Figure 8e* and *Video 2*) resolves the enrichment of mGBP2 at the PV membrane resulting in a rough surface with elongated very bright features, that are sufficiently large to be resolved by far field confocal microscopy.

*Figure 10* provides a scheme derived from the observed mGBP interactions in living cells with molecular resolution at various stages after *T. gondii* infection. Our hetero-FRET data of MFIS measurements clearly reveal interactions of mGBP2 in multimers with itself, mGBP1, and, to a lesser extent, with mGBP3 but not with mGBP6. However, the interplay between mGBP2 and mGBP5 is different. The two proteins can be coprecipitated (*Figure 4—figure supplement 2*), but the complex shows no FRET (*Figures 4* and *6*). Given the experimentally achieved concentrations in the cytosol and the corresponding enrichment in the VLS, the observation that fluorescence anisotropy of G-mGBP2 increased while its donor lifetime remained unchanged suggests either an interaction of mGBP2 and mGBP5 via adaptor molecules, so that they are not in close proximity and hence FRET inactive, or the rather unlikely case of an unfavorable static orientation of the fluorophores. It is noteworthy that, upon infection, oligomerization and accumulation of the mGBPs in VLS is reversible, so that the VLS serve as protein reservoir to accomplish a fast attack of the parasite after infection.

Both mGBP1 and mGBP2 have been implicated in *T. gondii* defense in single gene deficient mice (*Degrandi et al., 2013*; *Selleck et al., 2013*). Since mGBP1 still recruits to *T. gondii* in mGBP2$^{-/-}$ cells (*Degrandi et al., 2013*), the high level of colocalization and interaction between mGBP1 and mGBP2 and their important roles in *T. gondii* control strongly argue for a cooperative effect at the PVM of *T. gondii*. Interestingly, reconstitution of mGBP2 in mGBP$^{chr3}$-deleted MEFs did not allow a sufficient control of *T. gondii* replication, while reconstitution of mGBP1 partially restored the WT phenotype (*Yamamoto et al., 2012*). Although more studies on the hierarchy of mGBPs are needed to fully understand the individual roles of each GTPase, this might hint that mGBP2 acts in concert with mGBP1 and possibly other mGBPs to exert its molecular antiparasitic activity.

The dissociation constant $K_{D,oligo}$ of mGBP2/mGBP3 heteromers is 25 times larger than that of mGBP2/mGBP2 homomers. Thus, it is not surprising that mGBP3 colocalized only partially in the same VLS (*Figure 3*, *Figure 10*). Strikingly, mGBP6, which also localizes in VLS and recruits to the PVM of *T. gondii*, is predominantly found in different VLS and shows no interaction with mGBP2 by FRET and co-IP. The different localizations of mGBP multimers argue for distinct individual functional roles in *T. gondii* immunity to be elucidated in the future.

Recently, an essential function for the cassette of autophagy proteins, including Atg7, Atg3, and the Atg12-Atg5-Atg16L1 complex was demonstrated in cellular anti-*T. gondii* immunity by facilitating IRG and GBP recruitment to the PVM (*Ohshima et al., 2014*; *Choi et al., 2014*; *Haldar et al., 2014*). This function appears to be independent of the classical autophagy degradation pathway (*Zhao et al., 2008*), but rather to play a role in the delivery of effectors to pathogen containing vacuoles (*Selleck et al., 2013*). Performing live cell imaging and MFIS analysis it could be shown that mGBP2 is loaded on the PVM of *T. gondii* within a few minutes post-infection, assembling a machinery of supramolecular complexes with mGBP1 and mGBP3.

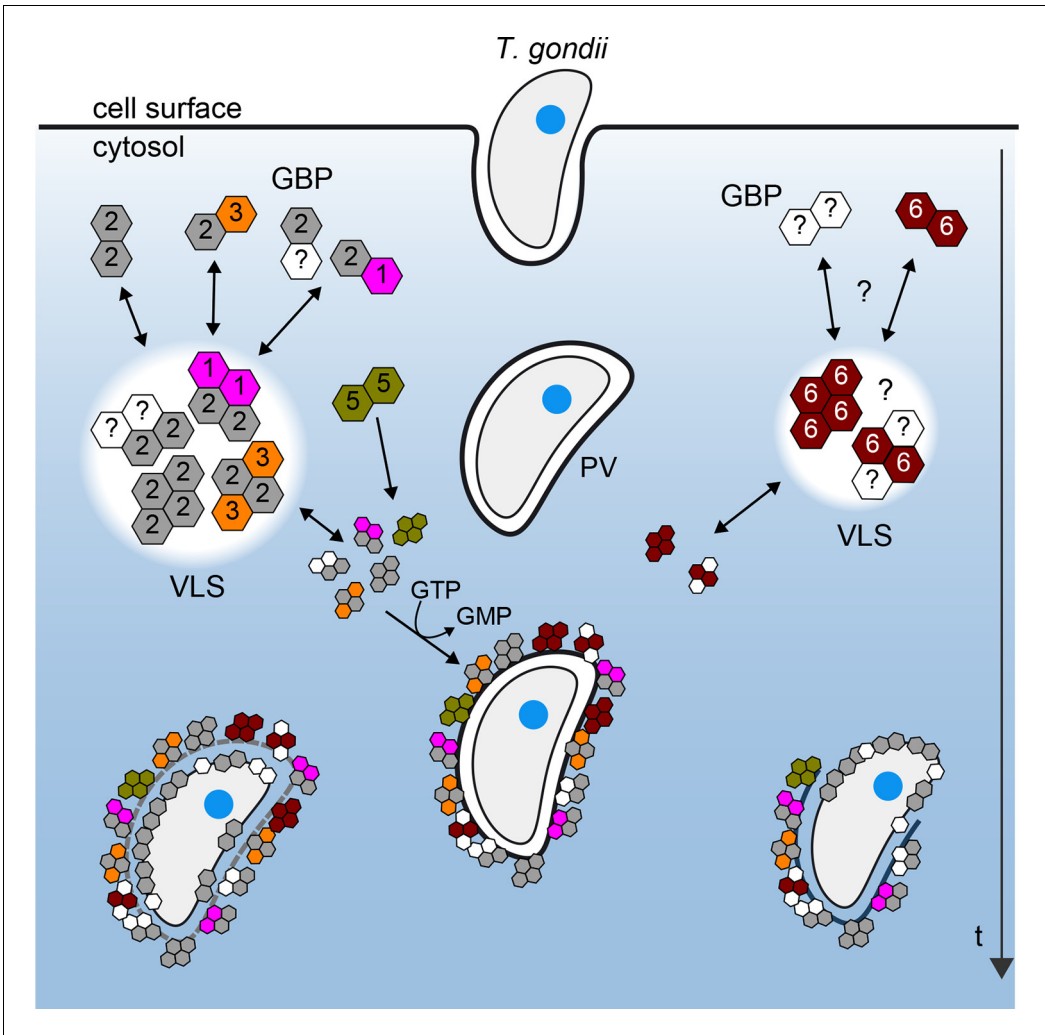

**Figure 10.** Schematic view of mGBP dynamics and multimerization in *T. gondii* infected cells. For details see Results and Discussion

It has been recently shown, that mGBP and IRG host proteins cooperate in destruction of PVs of *T. gondii* (*Haldar et al., 2015*; *2013*; *Yamamoto et al., 2012*). Previous studies in astrocytes and macrophages infected with type II *T. gondii* strains have shown that IRGs participate in mediating vesiculation of the PVM, resulting in the exposure of disrupted parasites to the host cytosol (*Ling et al., 2006*; *Martens et al., 2005*; *Melzer et al., 2008*; *Zhao et al., 2009a*; *Zhao et al., 2009b*). However, no colocalization of IRG proteins with the parasite plasma membrane has been reported previously. Here, we unambiguously show that mGBP2 directly targets the membrane of the parasite after penetration or disruption of the PVM.

Interestingly, GBP proteins, especially mGBP2, were shown to stimulate caspase-11-dependent pyroptosis in macrophages infected with Gram-negative bacteria which reside in vacuoles. There, GBP dependent induction of lysis of the pathogen-containing vacuoles and release of cytoplasmic LPS leads to the activation of the noncanonical inflammasome (*Pilla et al., 2014*; *Meunier et al., 2014*). Strikingly, a novel study suggests a direct bacteriolytic function of mGBPs, releasing patho-gen-associated molecular patterns into the cytosol, resulting in activation of the AIM2 inflammasome (*Man et al., 2015*; *Meunier et al., 2015*). Thus, based on our observations, it is likely that mGBPs promote not only disruption of the PVM, but also directly induce lysis of the plasma membrane of *T. gondii*. The hierarchy of events which might be involved in *T. gondii* targeting and elimination, such

as autophagic degradation (*Choi et al., 2014*) and/or inflammasome activation (*Ewald et al., 2014*; *Gorfu et al., 2014*; *Meunier et al., 2014*; *2015*), have yet to be determined.

These studies define mGBP2 as an important effector molecule of innate immunity in the host parasite interaction with apicomplexan parasites such as *T. gondii*, by providing seminal insight into its supramolecular assembly and cellular function. Further studies will be performed to address the question how this information can be exploited for anti-parasitic therapy.

## Materials and methods

### Expression constructs

The WT ORF of mGBP2 (NCBI accession numbers: mGBP-2, NM_010260.1) was subjected to site directed mutagenesis (QuikChange II Mutagenesis kit, Agilent, Germany) for generation of GTPase mutants R48A, K51A, E99A and D182N (*Kravets et al., 2012*) and isoprenylation mutant C586S (*Degrandi et al., 2013*) in the pEGFP-C2 plasmid (Clontech, France). The respective genes were then cloned into the pWPXL plasmid (Trono lab, Switzerland) as N-terminal GFP-fusion constructs. For the cloning of mCherry constructs, the pWPXL plasmid was modified by exchanging of the gene for GFP by the gene for mCherry. The ORFs of mGBP1 (NM_010259.2), mGBP2, mGBP3 (NM_001289492.1), mGBP5 (NM_153564.2), mGBP6 (NM_194336.2) were then cloned into the modified pWPXL plasmid as N-terminal mCherry-fusion constructs. The lentiviral envelope vector pLP/VSVG (Invitrogen, Germany) and the packaging vector psPAX2 (Trono lab) were used for the lentiviral genetic transfer.

### Cell culture and transduction

MEFs were cultured in Dulbecco's modified Eagle's medium (DMEM, Invitrogen/Gibco, Germany) supplemented with 10% (v/v) heat-inactivated low endotoxin fetal bovine serum (FBS, Cambrex, Germany), 100 U/ml penicillin, 100 µg/ml streptomycin, 2 mM L-glutamine (Biochrom, Germany) and 0.05 mM β-mercaptoethanol (Invitrogen/Gibco). Human foreskin fibroblasts (HS27, ATCC CRL-1634) were hold in culture in Iscove's Modified Dulbecco's Medium (IMDM, Invitrogen/Gibco) with the same supplementations. 293FT cells were cultivated in DMEM supplemented with 10% FBS, 100 U/ml penicillin, and 100 µg/ml streptomycin. All recombinant lentiviruses were produced by transient transfection of 293FT cells according to standard protocols. Briefly, subconfluent 293FT cells were cotransfected with 20 µg of a plasmid vector, 10 µg of psPAX2, and 5 µg of pLP/VSVG by calcium chloride precipitation in FBS free medium. After 6 hr medium was changed (10% FBS), and supernatants with recombinant lentivirus vectors were harvested 48 hr later. Alternatively, the trasfection was performed utilizing the jetPRIME trasfection reagent (Polyplus, New York, NY) in medium supplemented with 10% FBS. MEFs were seeded in 24 well plates (Corning , Germany) and transduced with 600 µl of lentivirus with 25 µg Polybrene (Merck Millipore, Germany). After 4 hr of incubation medium was changed. The transduction efficacy was analyzed by flowcytometry. Subsequently, GFP or GFP/mCherry positive cells were sorted and cultivated.

Tachyzoites from *T. gondii* strain ME49 were maintained by serial passage in confluent monolayers of HS27 cells. After infection of fibroblasts, parasites were harvested and passaged as described previously (*Degrandi et al., 2007*).

### Infection of murine MEFs with *T. gondii*

Cells were stimulated with 200 U/mL IFNγ (R&D Systems, Minneapolis, MN) 16 hr before infection. For immunofluorescence, MEFs were cultured in 24-well plates (Falcon, BD Biosciences, Germany) on cover slips (ø 13 mm, VWR International, Germany) and inoculated with freshly harvested *T. gondii* at a ratio of 50:1. To remove extracellular parasites, cells were washed with PBS.

### Immunofluorescence analysis

Cells were fixed in 4% paraformaldehyde (PFA, Sigma-Aldrich, Germany) permeabilized with 0.02% saponin (Calbiochem-Merck)and blocked in 0.002% saponin with 2% goat serum (Biozol, Germany). The outer membrane of *T. gondii* was visualized by anti-SAG1 (Abcam, UK) at a concentration of 1/700. As secondary reagents, 1/200 concentrated Cy2-conjugated goat anti-rabbit IgG and Cy3-conjugated goat anti-mouse IgG plus IgM (Jackson ImmunoResearch Laboratories, UK) were used.

Nuclei were counterstained with 1/2500 4',6-diamidino-2-phenylindole (DAPI, Invitrogen). The cover slips were fixed in fluorescence mounting medium (Fluoromount-G, Southern Biotechnology Associates, Birmingham, AL). Fluorescence was visualized using a LSM780 confocal microscope (Zeiss, Germany). Image analysis and processing was performed with the ZEN (Zeiss) and Imaris (Bitplane, Switzerland) software.

### Confocal live cell imaging

Live cell imaging was performed using an LSM780 confocal microscope (Zeiss) at 37°C with 8% $CO_2$ and humidity saturated air. Cells were cultured and imaged on imaging dishes CG (MoBiTec, Germany) with Phenol-free cell culture media. Image analysis was performed with the software ZEN (Zeiss), Imaris (Bitplane) and AutoquantX3 (MediaCy, Rockwell, MD/Bitplane).

### MFIS setup

MFIS experiments (*Kudryavtsev et al., 2007*; *Weidtkamp-Peters et al., 2009*) were performed with a confocal laser scanning microscope (Olympus FV1000, IX81 inverted microscope) additionally equipped with a single photon counting device with picosecond time-resolution (PicoQuant Hydra Harp 400, PicoQuant, Germany). GFP was excited at 485 nm with a linearly polarized, pulsed (32 MHz) diode laser (LDH-D-C-485) at 0.4 µW at the objective (60x water immersion, Olympus UPlan-SApo NA 1.2, diffraction limited focus). mCherry was excited at 559 nm with a continuous-wave laser (FV1000) at 0.54 µW at the objective. The emitted light was collected in the same objective and separated into its perpendicular and parallel polarization (PBS 101, Thorlabs, Germany). GFP fluorescence was then detected by SPADs (PD5CTC, Micro Photon Devices, Italy) in a narrow range of its emission spectrum (bandpass filter: HC520/35 (AHF, Germany)). mCherry fluorescence was detected by HPDs (HPMC-100-40, Becker&Hickl, Germany), of which the detection wavelength range was set by the bandpass filters (HC 607/70, AHF). Images were taken with 20 µs pixel time and a resolution of 276 nm/pixel. With 485nm excitation, series of 40–100 frames were merged to one image and further analyzed with custom-designed software (*Widengren et al., 2006*) and at the web page (http://www.mpc.hhu.de/software/software-package.html).

### Pixel-wise MFIS analysis of fluorescence parameters

From the recorded GFP ($S_G$) and mCherry ($S_R$) signal intensities, background intensities of the regions where no cells localize were subtracted to determine fluorescence intensities of GFP ($F_G$) and mCherry ($F_R$) respectively. To determine fluorescence anisotropy ($r_D$) and fluorescence-weighted donor lifetimes ($<\tau_D>_f$) in each pixel, the histograms presenting the decay of fluorescence intensity after the excitation pulse were built with 256 bins and 128 ps per bin. The fitting procedures were described previously (*Stahl et al., 2013*; *Kravets et al., 2012*).

### Formal pixel-integrated MFIS-FRET analysis

In each obtained MFIS image, pixels in the VLS and in the cytosol in uninfected cells, and pixels at the PVM and in the cytosol in infected cells were separately selected according to fluorescence photon number (*Figures 1a*, *2a*, *4a* and *6a*). Photons from each pixel selection were integrated to an intensity decay histogram with 1024 bins and 32 ps per bin. The pixel-integrated histograms were formally fitted to quantitatively determine FRET parameters. In the model, fluorescence decay of FRET sample ($f_{mix}(t)$) is the sum of FRET-quenched donor decay ($f_{(D,A)}(t)$) weighted by its species fraction $x_{FRET}$ and unquenched donor decay ($f_{(D,0)}(t)$) weighted by (1- $x_{FRET}$):

$$f(t) = (1 - x_{FRET}) \cdot f_{(D,0)}(t) + x_{FRET} \cdot f_{(D,A)}(t) \tag{1}$$

Here, $f_{(D,0)}(t)$ could be pre-determined from donor-only measurements using a bi-exponential fit model:

$$f_{(D,0)}(t) = \sum_m x_{D0}^{(m)} \cdot \exp(-t \cdot k_{D\theta}^{(m)}) \tag{2}$$

in which m=2 because fluorescent proteins in living cells usually show a bi-exponential decay (*Suhling et al., 2002*). Fit parameters in $f_{(D,0)}(t)$ include two normalized pre-exponential factors $x_{D0}^{(m)}$ ($\sum x_{D0}^{(m)} = 1$) and two decay rate constants, $k_{D0}^{(m)}$. These pre-determined parameters from donor-

only measurements were then set as global restraints. The quenched donor decay $f_{(D,A)}(t)$ in **Equation (1)** is given by:

$$f_{(D,A)}(t) = f_{(D,0)}(t) \cdot \exp(-t \cdot k_{FRET}) \tag{3}$$

where $k_{FRET}$ is the FRET rate constant. The fitted parameters in the 1-$k_{FRET}$ model (**Equations 1–3**) are $x_{FRET}$ and $k_{FRET}$. This formal analysis revealed that mGBPs exhibit distinct FRET features in different cellular compartments (**Figure 6—figure supplement 2**).

## $\varepsilon_{mix}(t)$ and $\varepsilon_{(D,A)}(t)$ diagrams

FRET-related donor quenching histogram ($\varepsilon_{mix}(t)$) was plotted to directly separate different molecular species and visualize FRET efficiency in the pixel-integrated data. $\varepsilon_{mix}(t)$ is calculated as the ratio between normalized fluorescence decay of FRET sample, $f_{mix}(t)$, and of donor-only sample, $f_{(D,0)}(t)$:

$$\varepsilon_{mix}(t) = \frac{f_{mix}(t)}{f_{(D,0)}(t)} = x_{FRET}\,\varepsilon_{(D,A)}(t) + (1 - x_{FRET}) \tag{4}$$

The drop on a $\varepsilon_{mix}(t)$ diagram represents the species fraction of FRET-active complex, $x_{FRET}$.

In **Equation (4)**, $\varepsilon_{(D,A)}(t)$ is the ratio between $f_{(D,A)}(t)$ (**Equation 3**) and $f_{(D,0)}(t)$ (**Equation 2**) and describes the time-dependent occurrence of the FRET process.

$$\varepsilon_{(D,A)}(t) = \frac{f_{(D,A)}(t)}{f_{(D,0)}(t)} = \exp(-t \cdot k_{FRET}) \tag{5}$$

To directly compare different experiments, $\varepsilon_{(D,A)}(t)$ diagrams were plotted in **Figure 4d**. A steeper slope in $\varepsilon_{(D,A)}(t)$ diagram indicates that the experiment showed higher $k_{FRET}$.

## Pattern based pixel-integrated MFIS-FRET analysis

To resolve three characteristic protein species, namely mGBP monomer (with specie fraction $x_{mono}$), dimer ($x_{di}$) and oligomers ($x_{oligo}$) by analyzing time-resolved anisotropy $r_{mix}(t)$ (**Equation 6**) and time-resolved FRET-induced donor decay $\varepsilon_{mix}(t)$ (**Equation 7**) for homo- and hetero-FRET, respectively, both decays were fitted with a linear combination of three species-specific patterns.

*Homo-FRET.* The $r_{mix}(t)$ of homo-FRET data was fitted with:

$$r_{mix}(t) = r_0 \cdot \left( x_{mono} + x_{di}\left( \int p(k_{di})e^{-2\cdot k_{di}\cdot t}dk_{di} \right) + x_{oligo}\left( x_s e^{-2\cdot k_{oli,s}\cdot t} + x_l e^{-2\cdot k_{oli,l}\cdot t} \right) \right)e^{-t/\rho_{global}} \tag{6}$$

Here $p(k_{di})$ is the FRET-rate distribution of mGBP2 dimer complex as determined by the conformational sampling of the sterically allowed space (see Monte Carlo sampling of the donor-acceptor conformational space of mGBP2 dimer, **Figure 7b** and **Figure 7—figure supplement 1d**). $k_{olig,s}$ and $k_{olig,l}$ are formally assigned as the FRET rate constants of mGBPs oligomers of small and large sizes respectively (**Figure 7b**), and $x_s$ and $x_l$ are their normalized fractions. It has to be considered that energy can be transferred in forward and backward direction which doubles the rate constants. The monomer is described by a constant offset, because there is no FRET. The fundamental anisotropy $r_0$ for GFP molecules is known as 0.38. The global rotational correlation time $\rho_{global}$ was estimated as 15 ns given the molecular weight of G-mGBP2 fusion protein. Oligomer species which produced ultrafast decay components in $r_{mix}(t)$ resulted in a drop in the initial anisotropy (**Figure 6d**). With the knowledge of $r_0$ they can be determined together with other species in homo-FRET data.

*Hetero-FRET.* An analogous analysis was applied to the hetero-FRET data. The $\varepsilon_{mix}(t)$ (**Equation 4**) was fitted with:

$$\varepsilon_{mix}(t) = x_{mono} + x_{di}\left( \int p(k_{di})e^{-k_{di}\cdot t}dk_{di} \right) + x_{oligo,s}e^{-t\cdot k_{oli,s}} \tag{7}$$

in which $x_{oligo,s}$ denotes the species fraction of small oligomers. In the case of hetero-FRET, donor molecules in large oligomers (with species fraction $x_{oligo,l}$) could be strongly quenched by nearby acceptors up to nearly 100% and thus became irresolvable owing to the finite width of the instrument response function. Therefore the information of large oligomers in hetero-FRET data needed to be recovered according to the homo-FRET data. In the latter, the species fractions of small and

large oligomers were found equal in various compartments. Based on the relation $x_{oligo,s} = x_{oligo,l}$ the large oligomer fractions in hetero-FRET data were extrapolated. Moreover, such a coherent behavior between small and large oligomers indicated that they have a common origin; and the broad distribution of their rate constants showed that oligomers may consist of a variety number of units (*Figure 7b*). Hence, it is more meaningful to combine both oligomer species and generally sort protein species as monomer, dimer and oligomer as displayed in *Figure 6c*. The fits were performed by custom software programmed in MATLAB.

## Monte Carlo sampling of the donor-acceptor conformational space of mGBP2 dimer

Based on the hGBP1 crystal structure (*Prakash et al., 2000a*) homology models of the G-mGBP2 (PDB-ID: 1F5N, 4EUL) and mCherry-mGBP2 fusion protein (PDB-ID: 1F5N, 2H5Q) (*supplementary file 1a*) were constructed using MODELLER (*Fiser and Sali, 2003*). The homology models were protonated using PDB-ID 2PQR (*Dolinsky et al., 2007*). Then the protonated full-length protein models were mapped to a reduced representation solely consisting of the C-, $C_\alpha$-, N-, O- and the hydrogen atoms forming the NH-O bonds. The repulsion between the atom pairs (O, N), (C, O) and (C, N) were modeled as repulsive quadratic potential (*Kalinin et al., 2012*) and the existing hydrogen bonds as simple scaled attractive potential (1/r) preserving secondary structural elements. The sampling was performed on the $\phi$ and $\psi$ torsion angles. In each iteration step the torsion angle of one amino acid was changed by random value taken from a Gaussian-distribution with a width of 0.025 rad. The sampling of the conformational space was restricted to the linkage region. Thus, only the internal coordinates of the connecting linker were altered while the internal coordinates of the beta-barrels as well as the internal coordinated of the mGBP2 model were kept constant. Given the sampled conformation of the mCh-mGBP2 and the G-mGBP2 constructs a putative head-to-head dimer structures was created by superimposing the LG-domains onto the LG-domains in the dimer structure of hGBP1 in presence of GppNHp (PDB-ID: 2BC9) and discarding conformations with clashes (*Vopel et al., 2014*). To calculate the donor-acceptor distance, $R_{sim}$, in every simulated structure, on each fluorophore, two $C_\alpha$-atoms on the beta-barrel (Asn122 and Asn147 on GFP, Tyr125 and Glu149 on mCherry) were chosen (*Figure 7—figure supplement 1a*, *supplementary file 1a*), so that the connecting vector of the two atoms is a good approximation of the transition dipole. The distance between the middle points of the connecting vectors of the donor and acceptor is taken as the distance between the chromophores ($R_{DA,sim}$). *Supplementary file 1b* lists out the detailed calculation steps to determine the ($R_{DA,sim}$) and orientation factor ($\kappa^2$). For each simulated mGBP2 dimer conformation, its $k_{di}$ value was calculated according to:

$$k_{di} = (3/2) \cdot k^2 \cdot (1/\tau_{D(0)}) \cdot (R_0/R_{DA,sim})^6 \tag{8}$$

in which $\tau_{D(0)} = 1/k_0$ is 2.6 ns and the Förster radius ($R_0$) of GFP and mCherry is 52Å including $\kappa^2$ = 2/3. The donor-acceptor distance distribution obtained from the MC simulation of the mGBP2 dimer (*Figure 7—figure supplement 1e*, blue curve) was used as the prior input, and was further optimized according to experimental data measured in the cytosol using the maximum entropy method (MEM) (*Vinogradov and Wilson, 2000*). The optimized distance distribution (MEM-MC) is plotted in *Figure 7—figure supplement 1e* (red curve). The difference between both distributions is primarily in the short distance range, because a small fraction of oligomers is present in the experimental data (*Figure 7c*), but of course absent in the MC simulation of a dimer. The two distributions agree very well in the longer distance range, therefore the distribution from the MC dimer simulation ($k_{di}$) (*Figure 7b*) describes the experimental data in a valid manner.

## Determination of mGBP protein concentrations and binding curves

*mGBP protein concentrations*. The protein concentration is monitored via the fluorescence intensity of the fused fluorescent proteins. The detection volume of the MFIS microscope was calibrated by Fluorescence Correlation Microscopy (FCS) measurements of Rhodamine 110 (Rh110) to determine its shape and size. The fitting model applied to the obtained FCS curve assumes a 3-dimensional Gaussian-shaped volume, and a single diffusing species including transitions to a triplet state as shown in (*Weidtkamp-Peters et al., 2009*). From the Rh110 diffusion time of 32 μs and aspect ratio of 7, the detection volume $V_{det-GFP}$ of GFP was determined as 0.5 fl. The detection volume of

mCherry $V_{det\text{-}mCherry}$ is larger (0.8 fl) because of the longer wavelength. The brightness of GFP or mCherry in living cells was individually characterized from FCS measurements of freely diffusing fluorescent proteins in the cytosol. By fitting the same model function as in Rh110 experiment, it was found that with 0.54 µW of 559 nm laser excitation at the objective, mCherry brightness is $Q_{mCherry}$ = 0.1 kcpm (kilo counts per molecule) in the cytosol and that with 0.4 µW of 485 nm laser excitation, GFP brightness is $Q_{GFP}$ = 0.56 kcpm in the cytosol.

The average GFP fluorescence intensity of an image with GFP excitation was first corrected for detector dead time, and then the obtained intensity ($S_{G,G}^m$) was further corrected for quenching effect due to FRET:

$$S_{G,G}^u = \frac{S_{G,G}^m}{(1 - x_{FRET}) + x_{FRET} \cdot (1 - E)} \tag{9}$$

$S_{G,G}^u$ denotes unquenched GFP fluorescence intensity in the absence of hetero-FRET and was used to calculate the GFP concentration.

The average mCherry fluorescence intensity of an image with mCherry excitation was first corrected for detector dead time (**Becker, 2005**), and then used to calculate the mCherry concentration with the determined detection volume and the mCherry brightness.

The concentration of GFP ($C_{GFP}$) was determined by

$$C_{GFP} = \frac{S_{G,G}^u}{Q_{GFP} \cdot V_{det-GFP}} = \frac{S_{G,G}^u}{0.56 kcpm \cdot 0.5 fl} \tag{10}$$

The concentration of mCherry ($C_{mCherry}$) was determined by

$$C_{mCherry} = \frac{S_{R,R}}{Q_{mCherry} \cdot V_{det-mCherry}} = \frac{S_{R,R}}{0.1 kcpm \cdot 0.8 fl} \tag{11}$$

We note that we do not measure intensities of single-molecule events as described by Wu et al. (**Wu et al., 2009**) but intensity averages of pixel populations so that it is sufficient to use an average brightness $Q$ for the calculation of the fluorescent protein (FP) concentrations. In our pattern fits we usually find on average less than 10% of non-FRET species (**Figure 7c**). From this we conclude that under our conditions with one photon excitation of donors with low irradiance (as compared to the two photon excitation used by Wu et al. (**Wu et al., 2009**) and low FRET efficiency most of the mCherry molecules are active FRET-acceptors. The $K_{D,dim}$ of ~24 nM of mGBP2 dimerization determined in this way is very close to previous biochemical studies (**Kravets et al., 2012**).

In **Equation 9**, the FRET-active species fraction ($x_{FRET}$) is obtained from fitting of each measurement in pixel-integrated MFIS-FRET analysis using the 1-$k_{FRET}$ model (**Equations 1–3**). FRET efficiency, $E$, was calculated as:

$$E = 1 - \frac{\sum_m x_{D0}^{(m)} \cdot \left(k_{D0}^{(m)} + k_{FRET}\right)^{-1}}{\sum_m x_{D0}^{(m)} \cdot \left(k_{D0}^{(m)}\right)^{-1}} \tag{12}$$

Please refer to Formal pixel-integrated MFIS-FRET analysis for explanations on the symbols in **Equations 9 and 12**.

## Determination of dissociation constants

To quantify the dependence of the dimeric species fraction on the total protein concentration (initial increase, stationary phase followed by a decrease) the simplest possible model was used to approximate such a behavior. In this model the formation of a dimer and a subsequent formation of a tetramer formed by two dimers was assumed. The formation of a dimer and a tetramer can be described by two reactions with corresponding dissociation equilibrium constants:

$$A_1 + A_1 \leftrightarrows A_2 \quad K_{D,dim} = \frac{c(A_1)c(A_1)}{c(A_2)},$$

$$A_2 + A_2 \leftrightarrows A_4 \quad K_{D,oligo} = \frac{c(A_2)c(A_2)}{c(A_4)} \quad (13)$$

For given of equilibrium constants and a total protein concentration $c_T = c(A_1) + 2 \cdot c(A_2) + 4 \cdot c(A_4)$ the species concentrations $c(A_1)$, $c(A_2)$, $c(A_4)$ were determined numerically by solving the fourth polynomial equation $c_T(A_1)$ by a simple root-finding algorithm (*Ridders, 1979*) and minimize the disagreement between the modeled species fractions and the fitted fractions by a Quasi-Newton method (*Shanno and Kettler, 1970*). This model of stepwise oligomer formation was extended by the stepwise binding of dimer in a non-cooperative fashion (i.e. all equilibrium constants are equal to $K_{D,oligo}$) up to a dodecamer. If the total concentration of all oligomers (4–12) is used to display the binding isotherm, one obtains an only slightly broadened binding isotherm compared to the tetramer system. If this binding isotherm is fitted with the simpler tetramer model, a binding constant for dimer binding $K_{D,app-oligo}$ is obtained, which is slightly (factor 1.6) larger than the simulated value.

As no information on the cooperativity of binding and the spatially resolved GTP concentration was available, the formation of higher order oligomers was approximated by the minimal tetramer model for the following reasons: (1) FRET only senses its local environment (i.e. a limited oligomer size) thus the contribution of each monomer unit to the measured signal decreases with increasing oligomer size. (2) This simple model reduces the number of fitting parameters to an adequate level given the spread of the data-points. To conclude, a simple model with a Langmuir binding isotherm (i.e. non-cooperative binding) describes all experiments very well. The simulation showed that the obtained apparent dissociation constant $K_{Dapp,oligo}$ is a good approximation for the true $K_{D,oligo}$.

Note that the observed reduction in steady-state anisotropy ($r_D$) for cells of high mGBP2 concentration as displayed in *Figure 4c*, was mainly due to the large drop in the initial anisotropy of their time-resolved anisotropy ($r_D(t)$) as plotted in *Figure 6d*. Therefore the $K_{D,app}$ value (9 µM) derived from $r_D$ in fact reports the mGBP2 oligomerization processes that could produce such ultrafast depolarizing effect, and is very close to the 8 µM obtained by fitting $r_D(t)$ with the species-resolved model. Hence, the two independent approaches interrogating the same oligomerization process delivered very consistent results, verifying the reliability of the analyses.

## Maximum FRET rate constants

Due to its inverse sixth-power distance dependence (*Equation 14*), FRET depends on molecular proximity and cannot occur between remotely located fluorescent proteins. Consequently, in large mGBP oligomers, the FRET-induced donor quenching will eventually saturate regardless of the presence of more acceptors simply because they are too distant. If assuming that the mGBP proteins are arranged homogeneously in mGBP oligomers, the maximum $k_{FRET}$ can be estimated following the ideas of T. Förster (*Förster, 1949*).

Here, the case of a single donor is considered, the FRET rate constant $k_{FRET}$ from the donor to $N$ surrounding acceptors is given by *Equation 14* using the parameter in *Equation 8*.

$$k_{FRET,max} = \frac{1}{\tau_{D(0)}} \sum_{k=1}^{N} \left( \frac{R_0}{R_{DA,k}} \right)^6 \quad (14)$$

with $R_{DA,k}$ being the distance between the donor and the $k$-th acceptor $k_{FRET}$. Assuming that the acceptors that attached on mGBPs are homogeneously distributed around the donor and closed packed with a minimum inter-fluorophore distance $R_{min}$, which is ~26 Å given by the molecular dimensions of fluorescent proteins, a similar estimation of the maximum $k_{FRET}$ as in (*Förster, 1949*) can be performed.

To determine the maximum FRET-rate constant at which a donor molecule is quenched by multiple acceptors it was assumed that at saturation protein concentrations the space around the donor is fully filled by acceptors and the space that is occupied by the donor cannot be occupied by the acceptor. If a donor is homogenously surrounded by acceptors, given a constant molecular density $\rho$ (number of acceptors per volume), which are separated at least by a distance of $R_{min}$ from the donor, the FRET-rate constant is given by:

$$k_{FRET} = \frac{1}{\tau_0} \int_{R_{min}}^{\infty} \rho \cdot 4\pi R^2 \left(\frac{R_0}{R}\right)^6 dR = \rho \frac{R_0^6}{\tau_0} \frac{4}{3}\pi \cdot \frac{1}{R_{min}^3}$$

$$= \frac{1}{\tau_0} \frac{R_0^6}{R_{min}^3 \cdot R_{mol}^3} \tag{15}$$

$R_{mol}$ is the mean radius of the acceptor molecules and relates to molecular density $\rho$. Given the molecular structure of mCherry in mCh-mGBPs fusion proteins, $R_{mol}$ is approximated by 31 Å. The minimum possible distance $R_{min}$ is given by the structure of the fluorescent proteins (~20–30 Å). Therefore, the maximum possible FRET-rate constant $k_{FRET}$ was approximated by ~15 ns$^{-1}$.

## Scanning fluorescence intensity distribution analysis (FIDA) for determination of oligomer size

To investigate the size (composition) of mGBP2 oligomer locating at the PVM which can exceed the detectable range of FRET technique (> 10 nm), FIDA from (*Kask et al., 2000*) was adapted for imaging measurements and employed in infected G-mGBP2 expressing cells. Given the recorded photon trace in the image line of selected PVM area, 20 μs binned new sliding with 2.5 μs (1/8 × pixel time) steps intensity traces were computed. Then a corresponding 2D matrix of green versus red photon counts from all the time windows is generated and analyzed by 2D FIDA. The average brightness, $<Q_{oligo}>$, and average number, $<N_{oligo}>$, of the mGBP2 oligomers could be determined. The average number of mGBP2 units (*Figure 7e*) per oligomer $<N_{mGBP2}>$ is calculated as the ratio of obtained $<Q_{oligo}>$ to single GFP brightness $Q_{GFP}$:

$$\langle N_{mGBP2}\rangle = \frac{\langle Q_{oligo}\rangle}{Q_{GFP}} \tag{16}$$

Based on these two average numbers of oligomers and mGBP2 units per pixel and knowing the excitation volume of the setup, the average mGBP2 concentration $<c_{mGBP2}>$ is calculated as

$$\langle c_{mGBP2}\rangle_{FIDA} = \frac{\langle N_{oligo}\rangle \langle N_{mGBP2}\rangle}{N_A \cdot V_{det}} \tag{17}$$

where $N_A = 6.022 \times 10^{23}$ $mol^{-1}$ is the Avogadro's number and $V_{det} = 0.5$ fl – excitation volume of the used laser. The mGBP2 concentration calculated from scanning FIDA was compared with that directly derived from the GFP intensity as a control. *Figure 7—figure supplement 1f* shows the good agreement between both methods.

## Acknowledgements

We thank Julia Hartmann and Karin Buchholz for excellent experimental assistance. We thank Oleg Opanasyuk for discussions on statistical error estimations in fluorescence decay analysis. QM and TOP thank the International Helmholtz Research School of Biophysics and Soft Matter (IHRS BioSoft) for funding. The work was supported by grants of the Deutsche Forschungsgemeinschaft to CAMS and KP and the Jürgen Manchot Foundation to KP. All authors declare no conflict of interests.

## Additional information

### Funding

| Funder | Grant reference number | Author |
|---|---|---|
| Deutsche Forschungsgemeinschaft | Research Grant PF259/8-1 | Elisabeth Kravets Daniel Degrandi Klaus Pfeffer |
| Jürgen Manchot Foundation | | Klaus Pfeffer |
| International Helmholtz Research School of Biophysics and Soft Matter | Graduate Student Fellowship | Qijun Ma Thomas-Otavio Peulen |

The funders had no role in study design, data collection and interpretation, or the decision to submit the work for publication.

## Author contributions
EK, DD, QM, VK, Acquisition of data, Analysis and interpretation of data, Drafting or revising the article; T-OP, SF, RK, Analysis and interpretation of data, Drafting or revising the article; SWP, Acquisition of data, Drafting or revising the article; CAMS, KP, Conception and design, Analysis and interpretation of data, Drafting or revising the article

## Author ORCIDs
Claus AM Seidel, ⓘ http://orcid.org/0000-0002-5171-149X

# Additional files

## Supplementary files
• Supplementary file 1. (a) Amino-acid sequence settings in the MC molecular simulation. The residues used to define the dipole of the chromophoric groups are indicated. (b) Calculations of donor-acceptor distances ($R_{sim}$) and orientation factors ($\kappa^2$) from each sampled conformation from MC molecular simulation of G-mGBP2/mCh-mGBP2 dimer in steps. See Experimental procedures and *Figure 7—figure supplement 1* for details.

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
