## [Decision Letter]

Thank you for submitting your work entitled "Guanylate binding proteins (GBPs) directly attack *T. gondii* via supramolecular complexes" for consideration by *eLife*. Your article has been reviewed by three peer reviewers, and the evaluation has been overseen by a Reviewing Editor and Detlef Weigel as the Senior Editor.

The reviewers have discussed the reviews with one another and the Reviewing Editor has drafted this decision to help you prepare a revised submission.

Two of the three individuals involved in review of your submission have agreed to reveal their identity: George Yap and Thorsten Wohland.

Summary:

Although the postulated role of interferon-inducible GTPases (p65 family-GBPs and p47 family- IRGs) in mediating intracellular destruction of microbial pathogens is now well accepted, the only infection system where the mechanism is well understood at the subcellular level, is in the *Toxoplasma gondii* system. Several studies have documented the trafficking of interferon-inducible GTPases from cytosolic stores to the pathogen vacuole membrane, usually referred to as the PVM, where the GTPases (both GBPs and IRGs) presumably act in concert, to effect vesicular disruption of the PVM. A less well-understood event that ensues is the physical breakage of the plasma membrane of the parasite itself, which has now been rendered exposed to the cytosol.

Here, Kravets et al. have used advanced microscopic techniques to visualize and quantify the dynamics of transitioning from monomeric, to dimeric to multimeric states of the murine GBP2 and many of its putative partners from the same GBP family. In particular, they used multiparameter fluorescence imaging spectroscopy (MFIS), a technique developed by the Seidel group, to obtain a combination of single molecule fluorescence parameters, including FRET, anisotropy, fluorescence lifetimes, and molecular brightness to measure the localization and aggregation states of GBPs before and during infection of fibroblasts by T. gondii. They show that both GTPase activity as well as isoprenylation is required for GBPs membrane binding and oligomerization, and that only mGBP1, -2, and -3 interact as dimers or oligomers but not mGBP6, localizing to vesicle-like structures. The interaction for mGBP2 and -5 is qualitatively different. While they show increased anisotropy, they show no FRET, but colocalization at the PVM, which the authors interpret as an interaction of the two proteins via an adaptor molecule. Importantly, the authors measure dissociation constants for mGBP2 with mGBP1, 2 and 3. The authors show that the mGBP oligomerization and localization to VLS is reversible and that they localize to the parasitophorous vacuole membrane (PVM) of T. gondii after infection. Intriguingly, GBP complexes on the PV reach extends of several thousand multimers, which results in the destruction of the PV membrane. Surprisingly, the authors also provide the first evidence that after PV rupture, GBP2 also attacks the parasite membrane directly.

Essential revisions:

Overall, this is a technical tour de force study. Although some of the interactions between GBPs were hinted at in previous studies, and therefore anticipated, the authors have analyzed them in unprecedented detail using the techniques (MFIS-FRET) that have so far not been applied to the cellular level in any system. Their results also show the limitations in the insight that can be gained from simple co-localization studies, and the need for state-of-the-art technology. Nevertheless, the following major concerns need to be addressed by additional experimentation and analysis.

First, additional data need to be presented to show that PVM was disrupted. The authors show that the parasite within the vacuole can be labelled even without apparent disruption of the vacuolar membrane (Figure 8). Therefore an additional assay confirming the disruption in Figure 8 would strengthen their message.

Second, the key and most important finding, presented only at the end of this Results section, is the observation that GBP2 is shown to associate with the parasite plasma membrane and this event presumably leads to the previously described breakage of the parasite membrane. Previous reports on parasite plasma membrane disruption by the Howard lab and the Yap lab should be discussed in this context. If fully documented and validated, this would resolve an important question in the field. Thus, it would be important to solidify this seemingly anecdotal and preliminary analysis of this observation, by documenting the frequency of this event and by documenting the outcome (presumably of loss of parasite membrane integrity).

Third, in keeping with the rest of the manuscript, it would also be important to perform parallel analysis (done at the level of VLS and PVM) of mGBP2 oligomerization and dynamics at the level of the parasite membrane.

Below are additional comments by the reviewers that they deemed to be major and must be addressed in a revision.

1) Discussion paragraph 3: The authors describe the different fates of GBP-targeted T. gondii (3 different outcomes). Has this been quantified? What’s the percentage of T. gondii that survive GBP targeting, how many acquire GBPs on the membrane w/o apparent loss of PV integrity and how many lose vacuolar integrity?

2) Figure 7: To formally confirm that the vacuole indeed lysis after GBP2-mCherry targeting, I would suggest to include Galectin-3 staining or an alternative assay (differential permeabilization). Or include the use of a fluorescently-tagged Gal-3.

Figure 1, Figure 2: p-vales are missing. Is the difference between WT/R48A, WT/E99A, WT/D182N significant?

3) Subsection “Colocalization and hetero-FRET studies of mGBPs”/Figure 4/Figure 6: Why do the authors show epsilon(DA) fits in the lower panel separately. With this fit they could also show fits to epsilon(mix) directly (Equations. 3 and 4) providing a better evaluation of the quality of the fits?

4) Regarding the determination of mCherry concentrations, oligomerization, and hetero-FRET measurements: How do non-fluorescent labels influence the measurements? Especially the red fluorescent proteins are known to be slow in maturation and are easy to bleach and thus have a large fraction in dark states. This should lead to a bias in the concentration measurements of mCherry. In addition, there are multiple dark states, which might or might not be FRET competent (Wu et al. Biophys. J, 2009) and therefore also influence strongly the results on hetero-FRET. The authors should discuss this issue.

5) Is there any experimental evidence for the FRET rate distribution p(kdi) in subsection “Pattern based pixel-integrated MFIS-FRET analysis”? How robust are the fits to changes in this distribution?

In Equation 14, please either derive the results or give a reference, where the derivation is given, as the basic assumptions differ from Foerster's.

6) Is it possible that the lack of FRET between mGBP2 and mGBP5 is a consequence of the structure of the complex with an unfavorable orientation of the fluorophores? And are the fluorophores still freely mobile in the mGBP2/mGBP5 as they are in the mGBP2/mGBP2 complex?

---

## [Author Response]

*First, additional data need to be presented to show that PVM was disrupted. The authors show that the parasite within the vacuole can be labelled even without apparent disruption of the vacuolar membrane (Figure 8). Therefore an additional assay confirming the disruption in Figure 8 would strengthen their message.*

Previously, we already detected several hours after infection a decreased amount of vesiculated *T. gondii* PVMs in mGBP2−/− MEFs compared to WT MEFs proving an important role of mGBP2 in the disruption of the PVM of intracellular T. gondii (Degrandi et al., 2013).

Additionally, we now show that cytosolic mCherry protein diffuses into the PV space after disruption of a GFP-mGBP2 labeled PVM. This is shown in the new supplementary Video S4. Furthermore, we have added alternative proof of PVM disruption via a differential permeabilization assay, which is described below.

*Second, the key and most important finding, presented only at the end of this Results section*,

*is the observation that GBP2 is shown to associate with the parasite plasma membrane and this event presumably leads to the previously described breakage of the parasite membrane. Previous reports on parasite plasma membrane disruption by the Howard lab and the Yap lab should be discussed in this context. If fully documented and validated, this would resolve an important question in the field. Thus, it would be important to solidify this seemingly anecdotal and preliminary analysis of this observation, by documenting the frequency of this event and by documenting the outcome (presumably of loss of parasite membrane integrity).*

Thank you for these valuable suggestions. We have introduced a new paragraph addressing the role of IRGs in PVM disruption.

“It has been recently shown, that mGBP and IRG host proteins cooperate in destruction of PVs of T. gondii (Haldar et al., 2015, Haldar et al., 2013, Yamamoto et al., 2012). Previous studies in astrocytes and macrophages infected with type II T. gondii strains have shown that IRGs participate in mediating vesiculation of the PVM, resulting in the exposure of disrupted parasites to the host cytosol (Ling et al., 2006, Martens et al., 2005, Melzer et al., 2008, Zhao et al., 2009a, Zhao et al., 2009b). However, no colocalization of IRG proteins with the parasite plasma membrane has been reported previously.”

Additionally, we have further analyzed and quantified the events following mGBP2 recruitment to the PVM in depth.

“Additionally, the events following mGBP2 recruitment to the PVM were documented and quantified. For this, IFNγ stimulated G-mGBP2 MEFs were infected with T. gondii for 6 hours, fixed and the plasma membrane of T. gondii was stained with anti-SAG1. To determine the precise localization of mGBP2 at this time point, intensity profiles of G-mGBP2 and Alexa633-SAG1 were determined encompassing the PVM, the plasma membrane of the parasite and the cytosol of the parasite (Figure 9). A total of 110 intracellular mGBP2-positive T. gondii PVs out of two independent experiments were evaluated. About 1.8% of the parasites acquire mGBP2 on the plasma membrane without apparent loss of PV integrity (Figure 9). For 37.1% of counted parasites disruption of PVM and direct targeting of mGBP2 to the plasma membrane of the parasite was observed (Figure 9). The remaining 61.1% revealed mGBP2 targeting at the PVM without apparent disruption or permeabilization and targeting of the parasite plasma membrane (Figure 9). Occasionally, after 6 hours of infection, parasites with very aberrant SAG1 localization, providing evidence that these parasites were already non-viable. In such cases G-mGBP2 fluorescence inside the cytosol of the parasite could be found, suggesting a loss of the membrane integrity of the parasite (Figure 9).”

*Third, in keeping with the rest of the manuscript, it would also be important to perform parallel analysis (done at the level of VLS and PVM) of mGBP2 oligomerization and dynamics at the level of the parasite membrane.*

This is a very important point. However, the resolution of the MFIS setup is currently not sufficient to reliably differentiate by MFIS between PVM and the plasma membrane of the parasite in living cells. To address this issue STED-MFIS technology would be required. To our knowledge no comparable setup is available worldwide. We plan to establish such a system in the future, but we believe that these questions are outside the scope of this study and the technical possibilities right now.

*Below are additional comments by the reviewers that they deemed to be major and must be addressed in a revision. 1) Discussion paragraph 3: The authors describe the different fates of GBP-targeted T. gondii (3 different outcomes). Has this been quantified? What’s the percentage of T. gondii that survive GBP targeting, how many acquire GBPs on the membrane w/o apparent loss of PV integrity and how many lose vacuolar integrity?*

Thank you for these very important suggestions. We have addressed these points in detail. Please refer to our response to the second essential revision point.

2) Figure 7: To formally confirm that the vacuole indeed lysis after GBP2-mCherry targeting, I would suggest to include Galectin-3 staining or an alternative assay (differential permeabilization). Or include the use of a fluorescently-tagged Gal-3.

To answer these issues directly, we have decided to perform a differential permeabilization assay with type I and type II strains of *T. gondii* (see Results section and Table 2). Additionally, we have monitored the influx of cytosolic mCherry protein into the PV space after PVM disruption (see first essential revision point and Video 4).

“As previously reported, a rapid colocalization of mGBP2 with the PV of T. gondii type II strain ME49 but not of T. gondii type I strain BK in IFN-γ–activated MEFs was observed (Degrandi et al., 2007). After infection with T. gondii ME49, selective permeabilization experiments revealed that immunofluorescence labeling of SAG1 at the T. gondii plasma membrane could be detected for mGBP2-positive PVMs in the absence of saponin. In contrast, after infection with the virulent BK T. gondii, almost no SAG1-labeled parasites could be detected (Table 2). Please note that after saponin permeabilization virtually all ME49 or BK parasites could be labeled with anti-SAG1. This shows that targeting of mGBP2 to the PVM promotes permeabilization or disruption of the PVM, allowing influx of proteins into the PV space.”

*Figure 1, Figure 2: p-vales are missing. Is the difference between WT/R48A, WT/E99A, WT/D182N significant?*

We have added the corresponding information on statistical significance to Figure 1 and Figure 2.

*3) Subsection “Colocalization and hetero-FRET studies of mGBPs”/Figure 4/Figure 6: Why do the authors show epsilon(DA) fits in the lower panel separately. With this fit they could also show fits to epsilon(mix) directly (Equations. 3 and 4) providing a better evaluation of the quality of the fits?*

Because FRET-induced donor decay *ε_mix_(t)* displays the interaction state of an ensemble of proteins, which includes both FRET-active and -inactive species. The former is visualized as the decay in the *ε_mix_(t)*, which is *x_FRET_**ε*_DA_(t)*, while the latter is the offset. The FRET efficiency *(E)* is reflected by the steepness of the decay slope: the steeper the decay, the higher the *E.* Separate display of the fitted *ε_DA_(t)* visually removes the interference from the offset due to FRET-inactive species. Only in this way different datasets measured in different cellular localizations/ different cells can be compared because different contributions of FRET-inactive proteins are excluded.

We have rewritten the paragraph accordingly addressing *ε_DA_(t)* for the first time in connection with Figure 4.

*4) Regarding the determination of mCherry concentrations, oligomerization, and hetero-FRET measurements: How do non-fluorescent labels influence the measurements? Especially the red fluorescent proteins are known to be slow in maturation and are easy to bleach and thus have a large fraction in dark states. This should lead to a bias in the concentration measurements of mCherry. In addition, there are multiple dark states, which might or might not be FRET competent (Wu et al. Biophys. J, 2009) and therefore also influence strongly the results on hetero-FRET. The authors should discuss this issue.*

Thank you for this question. In the Methods section (paragraph 12) we added a more detailed description of the equations, which were used to calculate the concentrations and discussed the mCherry problem as stated below.

"We note that we do not measure intensities of single-molecule events as described by Wu et al. (Wu et al., 2009) but intensity averages of pixel populations so that it is sufficient to use an average brightness Q for the calculation of the fluorescent protein (FP) concentrations. In our pattern fits we usually find on average less than 10% of non-FRET species (Figure 7). From this we conclude that under our conditions with one photon excitation of donors with low irradiance (as compared to the two photon excitation used by Wu et al. (Wu et al., 2009) and low FRET efficiency most of the mCherry molecules are active FRET-acceptors. The KD,dim of ~24 nM of mGBP2 dimerization determined in this way is very close to previous biochemical studies (Kravets et al., 2012).”

Even if there was a small systematic error of the computed FP concentrations, this would lead only to a systematic change in every estimated KD value, but the relative differences among mGBP dimers and oligomers, and the conclusions will remain unchanged.

Moreover, as discussed in the answer to question 5, the GFP linker effects must be properly accounted for; otherwise a quantitative analysis becomes impossible. There is a considerable fraction of DA constructs (see Figure 7—figure supplement 1d), where the FRET-rate constant is very small, either because the donor and acceptor are too far away from each other or the dipole orientation factor κ^[2]^ is close to zero. These orientation- and linker-effects are already included in our pattern analysis with distributions.

Our FRET studies of more than 20 different fusion-constructs forming complexes in different cell types (mouse fibroblasts (this work), various human cell lines, roots of *Arabidopsis thaliana*, leafs of Nicotiana benthamiana), revealed, for saturating conditions of complex formation or tandems, a maximal fraction of non-FRET xno-FRET of 30%, but usually xno-FRET was comparable to this work (i.e. xno-FRET ≤ 10% ).

*5) Is there any experimental evidence for the FRET rate distribution p(kdi) in subsection “Pattern based pixel-integrated MFIS-FRET analysis”? How robust are the fits to changes in this distribution?*

As the reviewers are certainly aware of the resolution that can be achieved by time-resolved fluorescence measurements is limited by the noise of the experimental fluorescence decays (Kollner and Wolfrum, 1992). To address the questions about the theoretical resolution limit of fluorescence decay measurements recorded by TCSPC we estimate a lower limit of the statistical variances for a given model realization at a given counted photon number using the Cramér–Rao inequality and the Fisher information matrix (FIM) (Peulen et al.).

Resolution limit of FRET. We consider a two component system with equal species fractions and FRET distances RDA,1 and RDA,2=RDA,1+ΔRDA as simplest model of a distance distribution. For such a system the shape of the donor fluorescence decay in presence of the acceptor is given by:

f(t,RDA,1,RDA,2)=e−kD[1+(R0RDA,1)6] t+e−kD[1+(R0RDA,2)6] tUsing this model we address the question how many photons have to be measured at least to determine RDA,1 and RDA,2 (or ΔRDA) with a given confidence and which is the distance RDA,lim any second distance RDA,lim+ΔRDA are indistinguishable.

Author response image 1.Statistical error estimates of a two distance model with a distance R_DA,1_ and a second distance R_DA,1_+ΔR_DA_; a Förster radius of 50 Å, a fluorescence lifetime of 4 ns and a time-window of 50 ns.(**A**) Relative errors of δ1 per one counted photon of distances R_DA,1_, R_DA,2_ and their difference ΔR_DA_. (**B**) Isolines δRDA,1=0.5(blue line), δRDA,2=0.5(green line) and δΔRDA=0.5(red line) for 10^[6]^ counted photons. Vertical lines indicate the limiting distances R_DA,lim_ for parameters R_DA,2_ (green) and ΔR_DA_ (red). (**C**) Limiting distances R_DA,lim_ for a given number of detected photons for parameters R_DA,2_ (green) and ΔR_DA_ (red).**DOI:**
http://dx.doi.org/10.7554/eLife.11479.028

Let us accept as the criterion of the reliable estimation of some parameter the condition then relative error of this parameter is below 0.5 (confidence of 95%). Author response image 1 B shows isolines δ=0.5 for RDA,1, RDA,2and ΔRDA in case if NC=106 photons were detected. These isolines partition parameter space into four regions. In the region (i) all three parameters are resolved. In the region (ii) distances RDA,1, RDA,2 can be reliably determined while the error of their difference δ(ΔRDA) increases above value 0.5. In the region (iii) only the shortest distance RDA,1 can be determined. This means that in the region (iii) the distance distribution is only partially resolved and species with the FRET distance RDA,2 cannot be distinguished from non-FRET species. Finally, in the region (iv) none of parameters satisfies the given criterion.

Isolines of relative errors δ(ΔRDA) and δ(RDA,2) reveal special values of distance RDA,1above which either the difference between two distances or the second longer distance cannot be resolved correspondingly. The dependence of these limiting distances RDA,lim on the total number of detected photons Nph is shown in Author response image 1 C . Remarkably, the maximum resolvable distances depend only weakly (nearly logarithmically) on the number of measured photons. To be able to estimate the couple of parameters RDA,1≈1.2R0 and ΔRDA≈0.5R0 roughly Nph=104 photons have to be detected.

Consequences for measurements in cells. This theoretical consideration demonstrates that a very large photon number has to be recorded to resolve distance distributions by fluorescence decay analysis. Being aware of the limitations of the fluorescence decay analysis we measured a set of mCherry-eGFP tandem constructs in live cells (Ma et al.). In those measurements two discrete FRET-rate constants were enough to formally describe the data. However, these FRET-rate constants gave no molecular insights into the system. Hence, we simulated the respective tandems by Monte-Carlo simulations as also presented in this manuscript for mGBPs. For all measured mCherry-GFP tandems we found that the distance distributions generated by Monte Carlo simulations (considering distance and orientation effects) agree very well with the measured fluorescence decays (Figure 12) and describe the length dependence on the linker.

Author response image 2.Comprehensive characterizations of FRET-FLIM data.Two representative experiments of donor-only (GFP) and FRET (GFP-mCherry) samples (left). Fitting the sub-ensemble fluorescence decay containing 3.96×106 photons of the GFP-mCherry experiment (in red) with 1-kFRET, 2-kFRET and kFRET-distribution models resulted in reduced *χ_r_^[2]^* = 1.41 (in gray), 1.03 (in dark cyan) and 1.08 (in orange), respectively. Pre-determined donor-decay parameters were set as global restraints in all fits: *x_D0_^[1]^* = 0.854, *x_D0_^[2]^* = 0.146, *τ_D0_^[1]^* = 2.747 ns, *τ_D0_^[1]^* = 1.526 ns. Parameters obtained from *1-k_FRET_* fit were *x_FRET_ = 0.303* and *k_FRET_ = 0.556 ns^-1^*; from 2-*k_FRET_* fit: *x_FRET_ = 0.392, x_FRET_*^[1]^ = 0.561, *x_FRET_*^[2]^ = 0.439, *k_FRET_*^[1]^= 0.225 ns^-1^ and *k_FRET_*^[2]^ = 1.765 ns^-1^; and from distribution fit: *x_FRET_ = 0.652*.**DOI:**
http://dx.doi.org/10.7554/eLife.11479.029

Previously, simple worm-like chain models have been applied to describe the average transfer-efficiency in FP-tandems connected by flexible peptide linkers (Evers et al., 2006b), and computational tools have been used to model FPs distance distribution (Pham et al., 2007). Fluorescent proteins are known to interact weakly. Thus, helper proteins are used to enforce interactions (Grunberg et al., 2013). Our live-cell measurements for more than 20 different fusion-constructs in different cell types (mouse fibroblasts (this work), various human cell lines, roots of *Arabidopsis thaliana*, leafs of Nicotiana benthamiana) are in good agreement with these results, as we best describe our data by distance distributions. In our measurements we do not find compelling evidence for strong interactions of FPs.

As already mentioned the photon count recorded in each experiment is rather limited. If individual pixel-integrated fluorescence intensity histograms of hetero-FRET samples are fitted formally, only one FRET rate constant can be extracted (Figure 6—figure supplement 2) and the no/low-FRET fraction (1−xFRET) is obtained. However, the physical meaning of the obtained parameters is limited and cannot reflect the fractions of the involved FRET-species and as the FRET-rate constants vary, the datasets cannot be analyzed by a single global FRET-rate constant.

Hence, similarly to our mCherry-GFP tandems (in preparation) (Ma et al.), we used the Monte Carlo (MC) simulations of the fluorescent protein linkers to obtain distance distribution obtained from of mGBP2 dimer (figure above, blue curve). We used this distribution as the prior knowledge to optimize it according to experimental data measured in the cytosol using the maximum entropy method (MEM) (Vinogradov and Wilson, 2000). The optimized distance distribution (MEM-MC) is plotted in red. The difference between both distributions is primarily in the short distance range because a small fraction of oligomers is present in the experimental data (Figure 7), but of course absent in the MC simulation of a dimer. The two distributions agree very well in the longer distance range, therefore the distribution from the MC dimer simulation (*p(kdi)*) describes the experimental data in a valid manner.

Given such a limited amount of photons the hetero-FRET data was globally analyzed and knowledge on p(kdi) as obtained from the MC molecular simulation was applied to describe our experimental results in a meaningful manner. Moreover, the Homo-FRET data shows mGBP multimerization. Hence, we consider our approach which combines prior knowledge on the linker behavior (Evers et al., 2006b, Grunberg et al., 2013, Pham et al., 2007) and experiments as the only reliable solution to resolve different mGBP species (monomer, dimer and oligomer).

In the Results section we clarify this dependency of the methods by adding the following information:

“The information content in the experimental fluorescence decays is restricted by their noise (Kollner and Wolfrum, 1992). Given the limited amount of photons of the pixel-integrated fluorescence intensity histograms, the pattern fit uses structural information of molecular simulations (Figure 7—figure supplement 1) to obtain population fractions of all species. The structural information is based on prior knowledge of the dimerization interface (Vopel et al., 2014) and on Monte Carlo simulations of the linkers connecting the fluorescent proteins to the GBPs (see Methods section 11) (Evers et al., 2006a, Pham et al., 2007). The obtained species fractions of mGBP2 monomers, homo- or hetero-dimers and oligomers are displayed in Figure 7.”

*In Equation 14, please either derive the results or give a reference, where the derivation is given, as the basic assumptions differ from Foerster's.*

We try to improve the description in our text. Förster uses either point like (Förster, 1949) dyes and continuum approaches (Förster, 1948) to describe concentration dependent FRET (Förster, 1949) or energy migration by a diffusion-like partial differential equation (Förster, 1948). As he uses continuum approaches and the integrals are taken from “zero” (point like dyes) the FRET-rate constant is not “capped off”. The shortest possible distance determines the maximum found FRET-rate constant. Therefore, his solutions of the “time-resolved fluorescence” (Anregungswahrscheinlichkeit) are not directly applicable for fluorescent proteins where the donor and acceptors are at least separated by ~16 Angström. We only wanted a rough estimate of the maximum FRET-rate constant. Hence, we assumed that a donor is surrounded by a homogenous acceptor density ρ (number of acceptors per volume) “outside” of the GFP-molecule:

kRET=1τ0∫RminRmaxρ4πR2(R0R)6dR=ρR06τ043π⋅(∫RminRmax(1R4)dR)=ρR06τ043π⋅[1Rmin3−1Rmax3]By integration of the whole space (Rmax→∞) we obtain a maximum FRET-rate constant kFRET,max.

We changed this accordingly in the main text to:

“To determine the maximum FRET-rate constant at which a donor molecule is quenched by multiple acceptors it was assumed that at saturation protein concentrations the space around the donor is fully filled by acceptors and the space that is occupied by the donor cannot be occupied by the acceptor. If a donor is homogenously surrounded by acceptors, given a constant molecular density ρ (number of acceptors per volume), which are separated at least by a distance of Rmin from the donor, the FRET-rate constant is given by:

kFRET=1τ0∫Rmin∞ρ⋅4πR2(R0R)6dR=ρR06τ043π⋅1Rmin3=1τ0R06Rmin3⋅Rmol3Rmol is the mean radius of the acceptor molecules and relates to molecular density ρ. Given the molecular structure of mCherry in mCh-mGBPs fusion protein, Rmol is approximated by 31 Å. The minimum possible distance Rmin is given by the structure of the fluorescent proteins (~20-30 Å). Therefore, the maximum possible FRET-rate constant kFRET was approximated by ~15 ns-1.”

*6) Is it possible that the lack of FRET between mGBP2 and mGBP5 is a consequence of the structure of the complex with an unfavorable orientation of the fluorophores?*

This is a very good point. We have added remarks to our Result and Discussion section and mentioned the limits of measurements.

*And are the fluorophores still freely mobile in the mGBP2/mGBP5 as they are in the mGBP2/mGBP2 complex?*

To analyze the mobility of the FPs, it is useful to analyze the steady state anisotropies depicted in Figure 4, Figure 6, Figure 4—figure supplement 1 and Figure 6—figure supplement 1. mGBP5 behaves very similar to the other mGBPs in the MFIS plots and the concentration dependence of rD, i.e. there is no experimental evidence why the FPs in the mGBP2/mGBP5 complex should be not mobile as well.